# Reconciling post-orogenic faulting, paleostress evolution and structural inheritance in the seismogenic Northern Apennines (Italy): Insights from the Monti Martani Fault System

Riccardo Asti[1], Selina Bonini[1], Giulio Viola[1], Gianluca Vignaroli[1]

[1] University of Bologna – Department of Biological, Geological and Environmental Sciences (BIGEA)

*Correspondence to*: Riccardo Asti (riccardo.asti2@unibo.it)

**Abstract.** Structural inheritance plays a significant role upon the evolution of fault systems in different tectonic settings. Both positive reactivation of pre-orogenic extensional faults and negative reactivation of syn-orogenic reverse faults during orogenic cycles have been extensively studied and documented. By contrast, only few studies have addressed the impact of structural inheritance in regions undergoing polyphasic tectonic histories. Here, we present the Monti Martani Fault System (MMFS) case study (Northern Apennines, Italy) as a representative example of a seismically active region where to investigate the role of inherited pre-orogenic structural features upon the post-orogenic tectonic evolution. We collected outcrop scale fault-slip data therefrom to constrain fault geometry and kinematics as inputs to paleostress analysis. Based on data from extensional faults that controlled the Plio-Quaternary evolution of the system, we propose that the MMFS does not consist of a c. 30 km long, L-shaped single normal fault, as previously proposed in the literature, but is instead formed by a set of several shorter NW-SE trending extensional faults arranged in an en-echelon style. Paleostress analysis yielded three distinct extension directions during the Plio-Quaternary post-orogenic extension, oriented NE-SW, NNE-SSW and NW-SE. We relate the first two directions to local orientation fluctuations of the regional stress field interacting with the moderately oblique inherited structural features, and the latter direction to a short-live orogen-parallel extensional event whose geodynamic causes remain unclear. We suggest that the NE-SW regional post-orogenic extension direction controls the orientation of most of the NW-SE striking extensional faults, while the morphostructural trend of the Monti Martani Ridge and of its boundaries with the surrounding Plio-Quaternary Medio Tiberino and Terni basins is controlled by the strike of the ~N-S and ~E-W pre-orogenic (Jurassic) inherited structural grain, rather than by the orientation of the post-orogenic extension direction. We also discuss the implications of these observations upon the seismotectonics of the MMFS. Our findings suggest that, in contrast to previous suggestions, the fault system cannot be classified as an active and capable structural feature.

## 1. Introduction

In areas affected by polyphasic deformation, structural inheritance exerts a significant influence upon stress distribution and deformation partitioning, thus controlling the balance between reactivation of pre-existing structures and rupture along newly formed faults (e.g., Viola et al., 2009; 2012; Autin et al., 2013; Brune, 2014; Scheiber & Viola, 2018; Wang et al., 2021; Hoddge et al., 2024). While the role of syn-orogenic structures is well documented for active and fossil belts (e.g., Tavarnelli et al., 2004; Curzi et al., 2020; Del Ventisette et al., 2021; Viola et al., 2022; Tavani et al., 2023), it remains poorly understood how inherited pre-orogenic structural features are geometrically and kinematically linked to the final architecture of the post-orogenic fault system. While field-based structural studies have generally shown that the spatial distribution and orientation of post-orogenic faults tend to closely resemble those of the pre-orogenic inheritance (e.g., Mercuri et al., 2024), insights from modelling on extended regions suggest, on the other hand, that complex patterns of newly formed faults may also form in response to the imposed stress field, overprinting and reworking the original, pre-orogenic structural framework (Autin et al., 2013; Brune, 2014; Wang et al., 2021). The way post-orogenic faults interact with the pre-orogenic setting has major implication for tectonically active regions, as fault geometry and its orientation to the background stress field impacts upon earthquake recurrence (e.g., Cowie et al., 2012) and Coulomb stress transfer during seismic events (e.g., Mildon et al., 2017; Galderisi & Galli, 2020). Thus, it is particularly important to understand and constrain the role of structural inheritance in tectonically active regions that, as a result of complex and polyphasic tectonic histories, have become saturated with inherited structural features (e.g., Viola et al., 2009, 2012; Scheiber & Viola, 2018; Hoddge et al., 2024). It is useful to be aware that the mechanical anisotropy defined by structural inheritance is not limited to pre-existing faults, particularly in polyphase tectonic regions. In such cases, it is more appropriate to refer to an "inherited structural grain", which can be defined as the summation of, e.g., stratigraphic, geometric, kinematic and tectonic features that, as a whole, express the bulk anisotropy of the crustal block being later further deformed.

The Apennines are an excellent example of a tectonically active region recording a series of tectonic events, spanning from pre-orogenic rifting, syn-orogenic folding and thrusting, to post-orogenic extension (e.g., Cosentino et al., 2010; Carminati et al., 2012; Cardello & Doglioni, 2015; Conti et al., 2020). Many studies have documented the inversion of inherited structures in the Apennines, during both the syn-orogenic and post-orogenic phases (e.g., Butler et al., 2004; Tavarnelli et al., 2004; Scisciani, 2009; Calamita et al., 2011; Di Domenica et al., 2012; Pace & Calamita, 2014; Curzi et al., 2020; Del Ventisette et al., 2021; Viola et al., 2022; Tavani et al., 2023). Accordingly, the current seismicity related to the post-orogenic extensional regime in the inner part of the belt is likely affected by the pre-existing structural framework (e.g., Chiarabba & Amato, 2003; Buttinelli et al., 2018; Barchi et al., 2021). Recent studies have even documented polyphasic reactivations of pre-orogenic extensional faults during both positive and negative inversion (Mercuri et al., 2024). Additionally, the Northern, Central and Southern Apennines represent former distinct paleogeographic domains juxtaposed by ~N-S structural features that, before the orogeny, represented first-order paleogeographic boundaries. These are oblique to the strike of most of the orogenic structures

(i.e. NW-SE; Fig. 1a) but are also oblique to the present-day NE-SW extensional stress field that affects the internal domains of the chain.

Such a complex structural framework and tectonic scenario is well exemplified by the Monti Martani Ridge (MMR) in the southernmost portion of the Northern Apennines (Fig. 1b), where a Mesozoic-Cenozoic sedimentary succession crops out forming a ~N-S structural ridge that is bounded by intermountain basins (Medio Tiberino and Terni basins) formed during post-orogenic extensional tectonics.

In this work, we present new field and structural data from the MMR, the Monti Martani Fault System (MMFS) and the associated Medio Tiberino and Terni basins. We performed structural analyses of exposed fault surfaces deforming both the Mesozoic units and the Plio-Quaternary deposits. Aiming at constraining the local post-orogenic stress field at the time of faulting, we carried out paleostress analysis on extensional faults. Our field-based structural geology approach allows us to discuss the role of pre-orogenic structural inheritance on the Plio-Quaternary structuring of the area and to explore the possible relationships between post-orogenic extensional faulting and the Plio-Quaternary continental units to draw inferences on the current state of fault activity and its potential for generating fault displacement hazard.

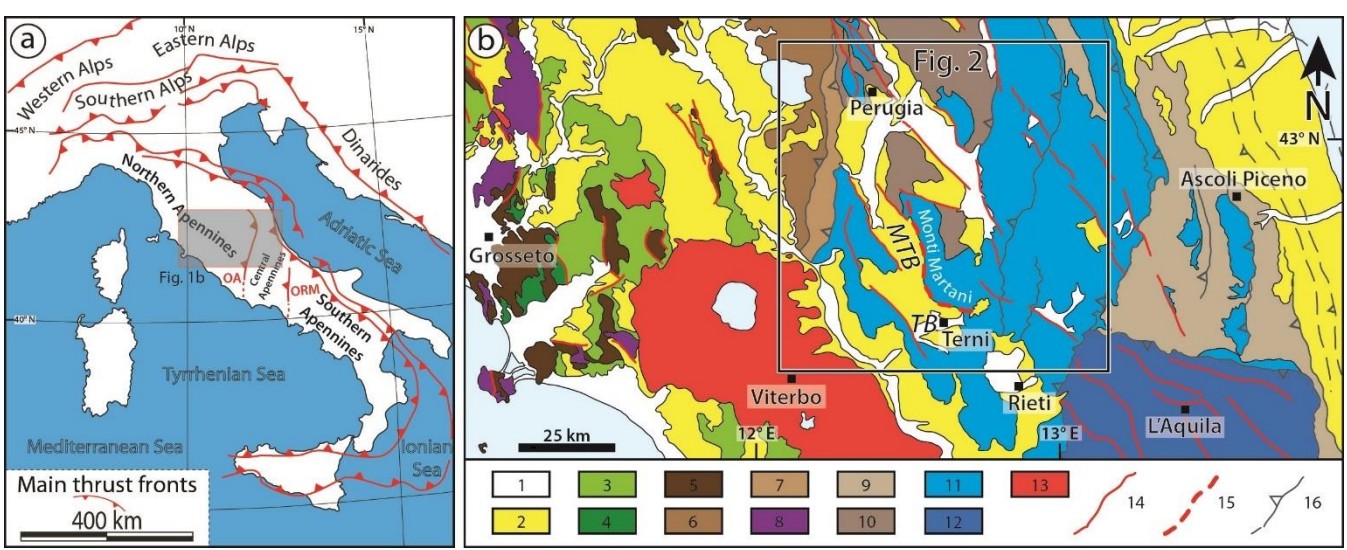

**Figure 1:** a) Schematic tectonic framework of the Italian peninsula with location of the main thrust fronts and subdivision of the main domains of the Apennines belt (modified after Zuccari et al., 2022). OA: Olevano-Antrodoco tectonic line; ORM: Ortona-Roccamonfina tectonic line. b) Tectonic map of the Northern Apennines at the transition with the Central Apennines. Redrawn and modified after Conti et al. (2020). 1: Quaternary deposits; 2: Miocene-Pleistocene succession; 3: Ligurian Units; 4: Subligurian Units, tectonic mélanges; 5: Tuscan Nappe; 6: Cervarola-Falterona Unit; 7: Rentella Unit; 8: Tuscan Metamorphic Units; 9: Umbria-Marche-Romagna Unit: turbidite successions of "minor" and outer basins; 10: Umbria-Marche-Romagna Unit: turbidite successions of the inner basins; 11: Umbria-Marche domain: Triassic-Miocene succession; 12: Lazio-Abruzzi carbonate Units; 13: Magmatic rocks; 14: Normal faults; 15: Martani Fault System; 16: Thrusts; MTB: Medio Tiberino Basin; TB: Terni Basin.

## 2. Geological setting

### 2.1. The Northern Apennines

The Northern Apennines of Italy (Fig. 1) are a seismically active belt that results from a complex and polyphase evolution that started in the Mesozoic. It recorded a Jurassic rifting phase related to opening of the Alpine Tethys, a later Late Cretaceous to Neogene orogenic phase related to Africa-Eurasia convergence and a final post-orogenic extensional phase that affected the internal part of the orogen since the Miocene (e.g., Boccaletti et al., 1971, 1985; Boccaletti & Guazzone, 1974; Principi & Treves, 1984; Vai & Martini, 2001; Molli, 2008; Conti et al., 2020 and references therein). Recent studies have also

documented several episodes of syn-sedimentary extensional tectonic activity between the end of the Tethyan rifting and the beginning of the orogenic phase in the Northern and Central Apennines (e.g., Centamore et al., 2007; Cipriani & Bottini, 2019a,b; Capotorti & Muraro, 2021, 2024; Sabbatino et al., 2021).

The Jurassic rifting started in the Sinemurian, resulting in the dismembering of an extensive carbonate platform and the progressive separation of Europe from the Adriatic plate (e.g., Bernoulli et al., 1979; Ziegler, 1988). On the Adria margin,

rifting led to the development of an articulated paleogeography with intrabasinal structural highs and lows leading to the deposition of pelagic successions in the structural lows and of condensed sequences on pelagic carbonate platforms (PCPs) on structural highs (e.g., Santantonio, 1993; Santantonio & Carminati, 2011; Cipriani et al., 2019, 2020; Santantonio et al., 2020). In the Northern and Central Apennines, particularly at their transition, the dominant strike of the Jurassic normal faults is ~N-S and ~ E-W (e.g., Coltorti & Bosellini, 1980; Calamita et al., 1991; Cipriani et al., 2020). This inherited structural template

was crucial for the localization of deformation during the subsequent orogenic phase through the positive inversion of the extensional pre-orogenic faults bounding the Jurassic structural highs (e.g., Butler et al., 2004; Pizzi & Galadini, 2009; Scisciani, 2009; Pace and Calamita, 2014; Scisciani et al., 2014; Tavani et al., 2023; Curzi et al., 2014).

Starting from the Late Cretaceous, Africa-Eurasia convergence led to the progressive closing of the Alpine Tethys, which culminated in the Eocene-Oligocene continental collision leading to the structuring of the Northern Apennines fold-and-thrust

belt (e.g., Doglioni et al., 1999; Rosenbaum et al., 2004; Mantovani et al., 2009 and references therein). This contractional phase caused the progressive stacking of nappes of different paleogeographic domains from the former Adriatic margin of the Ligurian-Piedmont Ocean (e.g., Elter, 1975; Vai & Martini, 2001; Vezzani et al., 2010; Cosentino et al., 2010; Conti et al., 2020). The eastward migration of NE-verging thrusting and folding (that is orthogonal to the orientation of the present-day belt axis) controlled the structural development of the belt toward the foreland. This resulted in overthrusting of oceanic and

transitional units, originally located farther west, on top of the external units (eastern side) belonging to the Adria microplate (Tuscan Domain, Umbria-Marche Domain).

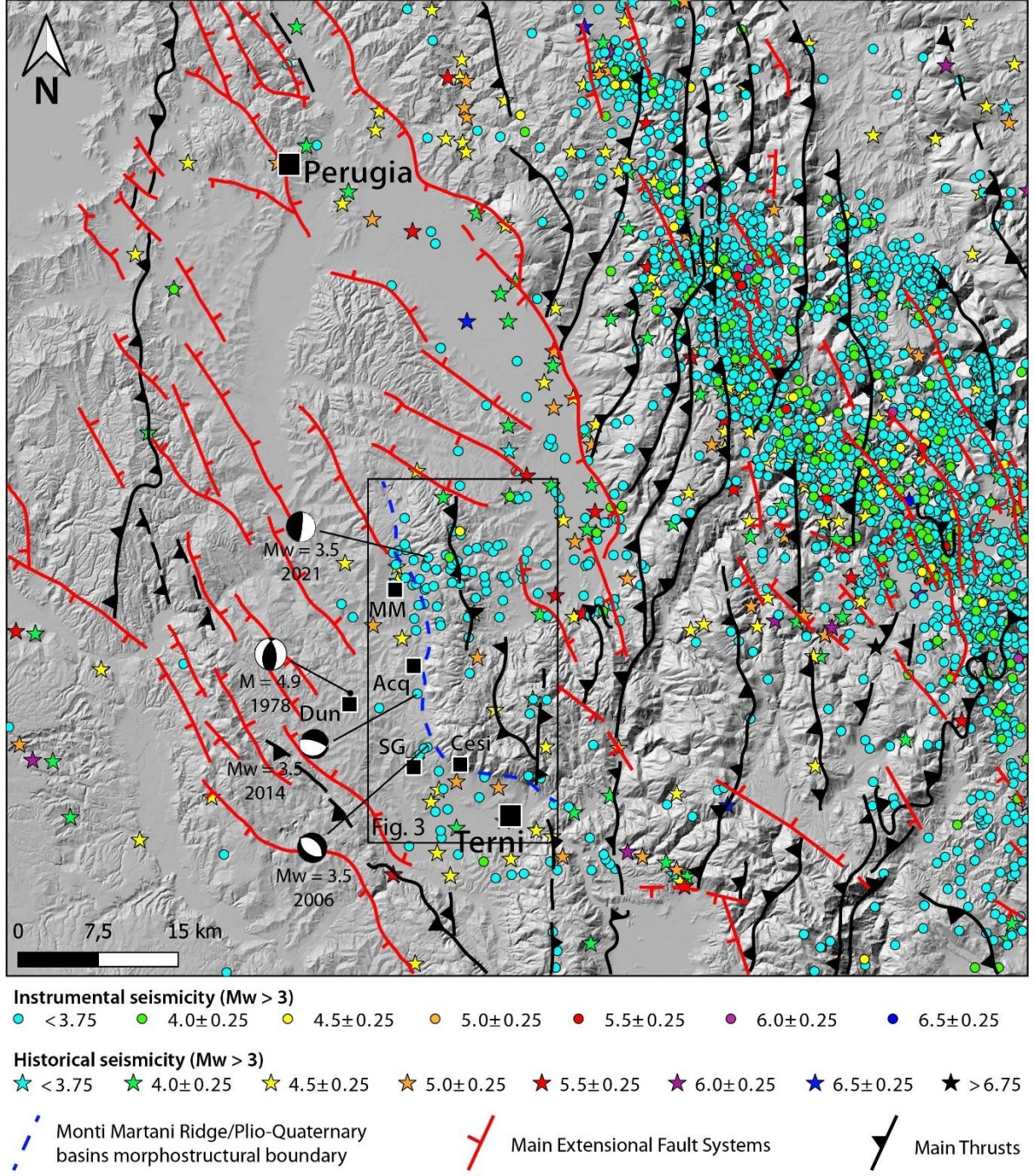

**Figure 2:** Digital Terrain Model of the Monti Martani Ridge and surrounding region with location of the epicenters of the instrumental (dots) and historical (stars) earthquakes with Mw > 3. Instrumental seismicity is from INGV-ISIDe catalogue (see ISIDe Working Group, 2007). Historical seismicity is from INGV-ASMI-CPTI catalogue (see Rovida et al., 2020, 2022). Earthquake focal mechanisms are from INGV-TDMT catalogue (see Scognamiglio et al., 2006), with the exception of the 1979 M = 4.9 Dunarobba earthquake which is from Gasperini et al. (1985). Main extensional fault systems and thrusts are from Calamita & Pierantoni (1995).

Starting in Miocene times, the internal part of Apennines underwent post-orogenic extension, resulting in the opening of the Tyrrhenian basin and of a series of intermontane, marine-to-continental basins on the Tyrrhenian side of the belt (e.g., Malinverno & Ryan, 1986; Keller et al., 1994; Cavinato & De Celles, 1999; Faccenna et al., 1997, 2001; Rosenbaum & Lister, 2004; Patacca et al., 2008; Carminati et al., 2012). The extensional front progressively migrated E- and NE-ward through the activation of NW-SE-striking normal faults. This tectonic regime still governs most of the current seismicity (producing up to Mw=7 events; ISIDe Working Group, 2007; Fig. 2) of the hinterland and axial regions of the Apennines (e.g., Cello et al., 1997; Mariucci et al., 1999; Boncio et al., 2000).

## 2.2. The Monti Martani Ridge

The MMR is a ~N-S trending, ~40 km long, L-shaped mountain ridge located in the southern part of the Northern Apennines. It separates the Medio Tiberino Basin to the W from the Umbria Valley Basin to the east (Figs. 1b, 2 and 3). To the south it is bounded by the Terni Basin. The ridge is composed of a Meso-Cenozoic carbonate platform transitioning upward to the pelagic deposits of the Umbria-Marche succession (sensu Centamore et al., 1986; Fig. 4), and a Miocene foreland basin succession (Accordi, 1966; Barchi, 1991; Brozzetti & Lavecchia, 1995; Fig. 3). The overall structural architecture of the ridge consists in a E-verging anticline (in the west), exposing the Meso-Cenozoic carbonate to marly succession and thrusting towards the ~E on a ~N-S trending syncline cored by Oligo-Miocene turbiditic units (Alfonsini et al., 1991; Calamita & Pierantoni, 1994, 1995; Alfonsini, 1995; Brozzetti & Lavecchia, 1995). On top of the Hettangian-Sinemurian platform carbonate (Calcare Massiccio Auctt.), the Jurassic stratigraphic succession reveals an articulated paleogeography. In fact, the typical pelagic to hemipelagic Jurassic units of the complete Umbria-Marche succession accumulated in the hanging wall basins and are locally replaced by the condensed succession of the Bugarone group (*Auctt.*), which deposited on top of Jurassic structural highs (Mariotti et al, 1979; Farinacci et al., 1981; Barchi et al., 1991; Brozzetti & Lavecchia, 1995; Bruni et al., 1995; Galluzzo & Santantonio, 2002; Fig. 4). Cipriani et al. (2020) therefore proposed that the Monti Martani area was a Jurassic pelagic carbonate platform, suggesting that the structural trends of the modern MMR (i.e. ~N-S and ~WNW-ESE) are directly inherited from the Jurassic rifting phase. Indeed, Jurassic faults have been documented to trend N-S, N20°, E-W and N110° (Bruni et al., 1995). Moreover, a significant influence of pre-Pliocene ~N-S and ~W-E trending structural features has been reported from the easternmost sector of the MMR (Montebibico area) and has been attributed to the inherited structural template of the Jurassic extension (Coltorti et al., 1995).

The present-day regional stress field of the internal part of the Northern and Central Apennines relates to an extensional tectonic regime with a minimum horizontal stress axis (Shmin) oriented NE-SW (Mariucci & Montone, 2024 and references therein). However, local deviations are reported from the Monti Martani area, where focal mechanisms of extensional earthquakes with a Shmin oriented N-S or E-W have been reported (Pondrelli et al., 2006).

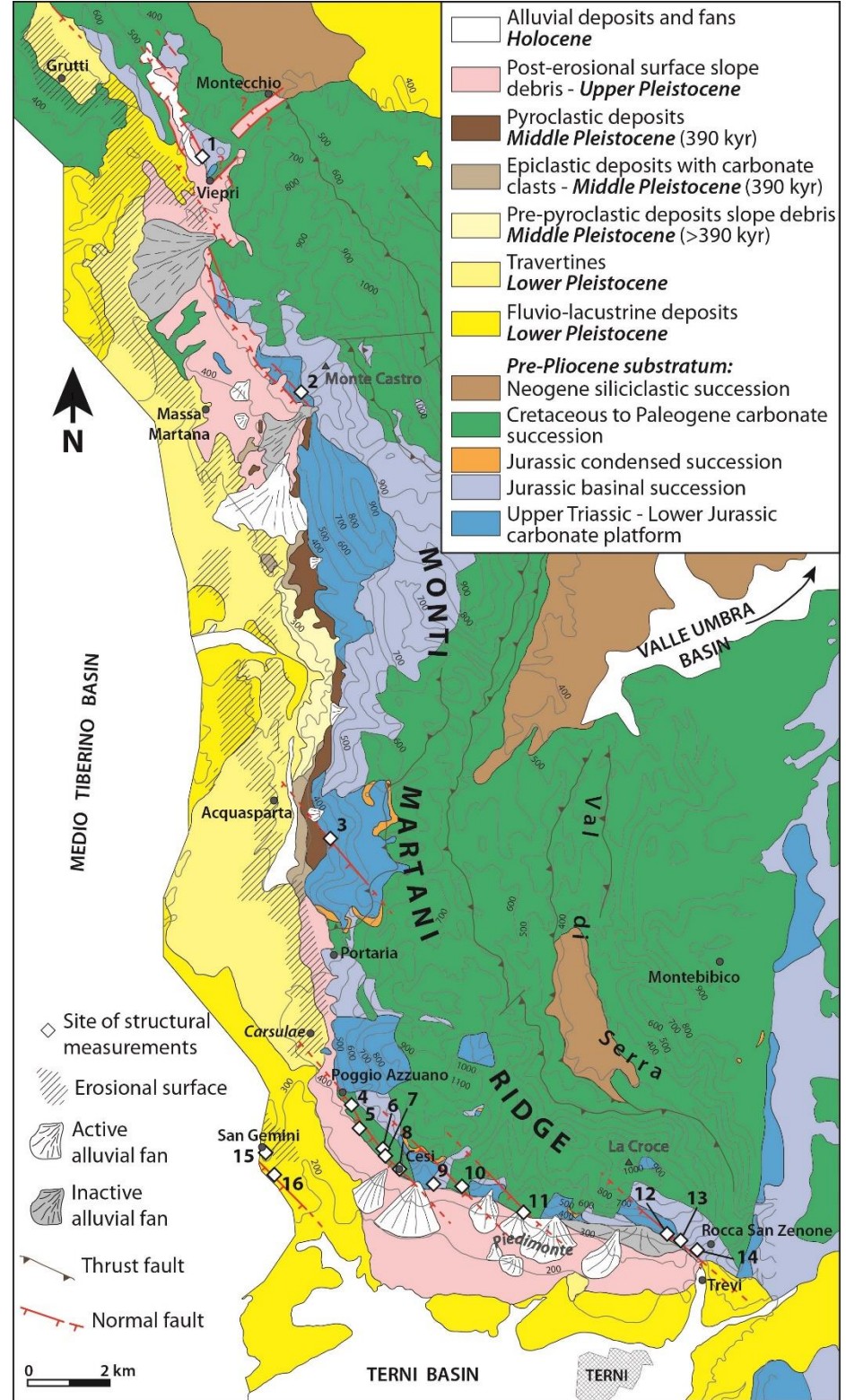

Legend:

- Alluvial deposits and fans *Holocene*
- Post-erosional surface slope debris - *Upper Pleistocene*
- Pyroclastic deposits *Middle Pleistocene* (390 kyr)
- Epiclastic deposits with carbonate clasts - *Middle Pleistocene* (390 kyr)
- Pre-pyroclastic deposits slope debris *Middle Pleistocene* (>390 kyr)
- Travertines *Lower Pleistocene*
- Fluvio-lacustrine deposits *Lower Pleistocene*

*Pre-Pliocene substratum:*
- Neogene siliciclastic succession
- Cretaceous to Paleogene carbonate succession
- Jurassic condensed succession
- Jurassic basinal succession
- Upper Triassic - Lower Jurassic carbonate platform

◇ Site of structural measurements

▨ Erosional surface

Active alluvial fan

Inactive alluvial fan

Thrust fault

Normal fault

0  2 km

**Figure 3:** Schematic geological map of the Martani Fault System (redrawn and modified after Bonini et al., 2003) with location of our structural stations. Traces of normal faults have been schematically redrawn according to original field observations presented in this study. However, the full structural complexity of the MMR (i.e., within the ridge) is not reported here. The reader can refer to the geological maps by Regione Umbria – Servizio Geologico (2013).

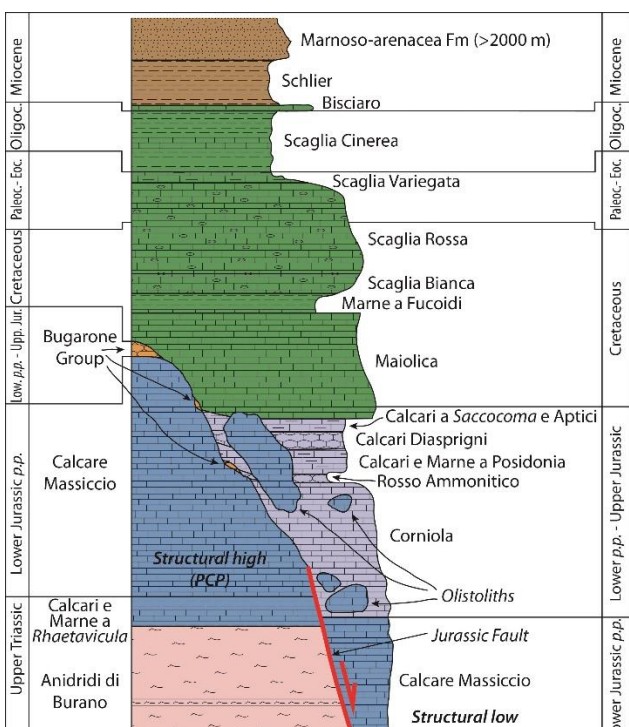

**Figure 4:** Chronostratigraphy and lithostratigraphic log of the Umbria-Marche succession (sensu Centamore et al., 1986) outcropping in the Monti Martani Ridge. Redrawn and modified after Curzi et al. (2024).

## 2.3. The Monti Martani Fault System

The MMR is bounded to the west and to the south by the MMFS (also referred to in the literature as Faglia Martana, Brozzetti & Stoppa, 1995, Faglia dei Monti Martani, Brozzetti & Lavecchia, 1995, or Martana Fault, Bonini et al., 2003), an extensional fault system separating the Meso-Cenozoic carbonate succession from the Plio-Quaternary Medio Tiberino (to the west) and Terni (to the south) basins. In the literature, this fault system is described as a single L-shaped structure, with an ENE-facing

concavity, including a ~ 30 km long N-S to NNW-SSE-striking segment extending from Grutti to San Gemini, and a ~10 km long WNW-ESE-striking segment extending from Cesi to Trevi (Brozzetti & Lavecchia, 1995; Bonini et al., 2003). Although this fault system is commonly represented in the literature as a continuous fault, with the two segments merging at the southwestern tip of the MMR, its surface expression is much more complex and ambiguous and does not necessarily match the morphostructure of the western and southern flanks of the MMR (see below). The cumulative vertical displacement across

the MMFS has been estimated to ~2000 m on a seismic cross-section across the northern tip of the MMR, even though the thickness of the associated Medio Tiberino Basin is only ~500 m (Barchi et al., 1991). Bonini (1998) proposed that the MMFS developed as an extensional reactivation of the ramp of an earlier Monti Martani thrust.

The Medio Tiberino Basin is a N-S to NNW-SSE trending graben to the west of the MMR and is filled by a ~500 m thick Upper Pliocene-Quaternary continental succession (Conti & Girotti, 1977, Ambrosetti et al., 1978, 1987a; Barchi et al., 1991; Basilici, 1997). The lower part of the succession consists of Upper Pliocene-Lower Pleistocene fluvio-lacustrine clay, sand and gravel, overlain by ~50 m thick Lower Pleistocene bedded travertine (Barchi et al., 1991). These deposits are locally overlain by slope debris and volcanic/volcanoclastic deposits (Brozzetti & Stoppa, 1995; Bonini et al., 2003) dated to the Middle Pleistocene by $^{39}Ar/^{40}Ar$ method (0.39 ±0.01 Ma; Laurenzi et al., 1994). This part of the succession is modelled by a planation surface which has been eroded during the Middle Pleistocene (Bonini et al., 2003) and has a smooth N-S topographic gradient. Reddish Upper Pleistocene slope debris postdates the erosional surface and crops out extensively along the western and southern slopes of the MMR (Bonini et al., 2003). The Terni Basin is a E-W trending graben exposed to the south of the MMR and is considered the southeastern branch of the Medio Tiberino Basin, with which it shares a common stratigraphic evolution (Basilici, 1993; Cattuto et al., 2002). To the north, the Terni Basin terminates against the ~ESE trending slopes of the southern termination of the MMR.

The Parametric Catalogue of Italian Earthquakes – CPTI (Rovida et al., 2020, 2022) contains few historical earthquake epicenters with Mw > 4 along the MMFS, with the strongest being the 5.1 Mw 1751 AD Terni earthquake and no historical earthquake reported in the N-S-striking segment of the fault system (Fig. 2). Instrumental seismicity recorded since 1985 (ISIDe Working Group, 2007) highlights moderate seismic activity (Mw ≤ 3.5) along the MMFS, with two main clusters located in the Massa Martana and Cesi/San Gemini sectors (Fig. 2). Only three focal mechanisms have been resolved along the MMFS for 3.5 Mw earthquakes (Scognamiglio et al., 2006) and they show inconsistent fault plane solutions: while the 2006 San Gemini earthquake yields an extensional mechanism with an horizontal NE-SW extension direction compatible with the current regional stress field, the 2014 Acquasparta earthquake yields ~N-S extension with a slightly oblique dextral component, and the 2021 Massa Martana earthquake is associated with a compressional mechanism with N-S striking planes and a W-plunging P axis (Fig. 2). Although the latter event seems totally incoherent with the overall regional stress field, it is quite consistent with the N-S striking planes with a sub-horizontal E-W trending P axis of the focal mechanism reconstructed for the 1978 Dunarobba earthquake, on the western margin of the Medio Tiberino Basin (Gasperini et al., 1985).

Despite the complexity of this seismic framework, only the ~ESE striking segment bounding the northern margin of the Terni Basin has been proposed as a potential seismogenic fault (Lavecchia et al., 2002). However, both segments of the MMFS are considered as active and capable faults in the ITHACA catalogue (ITaly HAzard from CApable faults; ITHACA Working Group, 2019). Accordingly, Bonini et al (2003) suggested that the MMFS cuts through Upper Pleistocene slope debris and displaces the archeological ruins of the Roman town of Carsulae. However, this interpretation has been challenged by later archeological studies (Aringoli et al., 2009; Bottari & Sepe, 2013).

## 3. Methods

In this work, we used a field-based approach to characterize fault geometry and kinematics and to make inferences on the relationships between faulting and deposition of the Plio-Quaternary basin infill. In particular, we focused our attention on mapping the faults exposed along the morphological boundary between the MMR and the Medio Tiberino and Terni basins (i.e., along the trace of the supposed L-shaped Martana Fault System; Fig. 3). We defined their orientation, geometry, kinematics and, when possible, cross-cutting relationships between fault populations. Most of the measurements were made in the Mesozoic carbonate succession of the MMR (i.e., in the footwall of the supposed L-shaped MMFS), while only few structural stations were studied in the Plio-Quaternary basin fill due to a substantial lack of exposures (Fig. 3; Table 1).

| Site n° | Locality | Latitude | Longitude | Data number | Data type |
|---------|----------|----------|-----------|-------------|-----------|
| 1 | Viepri | N 42° 49' 51.373'' | E 012° 31' 15.008'' | 38 | Faults + Veins |
| 2 | Massa Martana (Monte Castro) | N 42° 46' 52.559'' | E 012° 33' 02.951'' | 6 | Faults |
| 3 | Acquasparta (quarry) | N 42° 40' 24.271'' | E 012° 33' 25.502'' | 11 | Faults |
| 4 | Poggio Azzuano | N 42° 37' 24.635'' | E 012° 34' 12.472'' | 42 | Faults + S-C structures |
| 5 | Poggio Azzuano | N 42° 37' 09.607'' | E 012° 34' 20.333'' | 9 | Faults |
| 6 | Cesi | N 42° 36' 42.534'' | E 012° 35' 01.513'' | 20 | Faults |
| 7 | Cesi | N 42° 36' 39.509'' | E 012° 35' 02.279'' | 26 | Faults + S-C structures |
| 8 | Cesi (Grotta Eolia) | N 42° 36' 30.530'' | E 012° 35' 12.480'' | 27 | Faults |
| 9 | Cesi | N 42° 36' 19.106'' | E 012° 35' 51.923'' | 11 | Faults |
| 10 | Cesi (quarry) | N 42° 36' 18.339'' | E 012° 36' 09.296'' | 124 | Faults |
| 11 | Piedimonte (Madonna dell'Olivo) | N 42° 35' 59.432'' | E 012° 37' 23.148'' | 9 | Faults |
| 12 | La Croce | N 42° 35' 41.711'' | E 012° 39' 56.370'' | 12 | Faults |
| 13 | Fontana della Madonna | N 42° 35' 39.459'' | E 012° 40' 10.450'' | 8 | Faults |
| 14 | Rocca San Zanone | N 42° 35' 32.093'' | E 012° 40' 33.144'' | 6 | Faults |
| 15 | San Gemini | N 42° 36' 35.124'' | E 012° 32' 58.784'' | 17 | Fractures |
| 16 | San Gemini (quarry) | N 42° 36' 06.480'' | E 012° 33' 32.610'' | 11 | Faults |

**Table 1:** Location of structural stations with number and type of data collected

Statistical analysis of fault and striae populations in the cumulative fault dataset was carried out with the Daisy v.4.1 software (Salvini et al., 1999; Salvini, 2002; Fig. 5). Structural data were plotted for each structural station (Fig. 6) on Schmidt net, lower hemisphere stereographic projections drawn with the Win-Tensor program (Delvaux, 1993). When sufficiently abundant and statistically meaningful, these data were used to perform paleostress analysis to reconstruct the orientation of the paleostress tensor for each structural station (Fig. 7). Paleostress analysis was performed with the Win-Tensor program (Delvaux, 1993), which uses a refined version of the Right Dihedron method (Angelier & Mechler, 1977) and an iterative Rotational Optimisation procedure (Delvaux & Sperner, 2003). Data separation for tensorial analysis followed the workflow described in Mattila & Viola (2014) and was first performed based on field observations (i.e., cross-cutting relationships between faults or separation of different kind of structures, such as faults, veins, cleavage planes, etc.) and on a qualitative

kinematic analysis of the stereonets (i.e., definition of striae trend populations). We then adopted an iterative approach by
using in sequence the Right Dihedron method and the Rotational Optimisation procedure to compute a paleostress tensor while progressively removing those data that turned out to be kinematically incompatible with the calculated stress model. In particular, we removed all data with a slip deviation angle $\alpha > 30°$, which is defined as the acute angle between the measured slip vector on a fault plane and the theoretical optimal slip vector on the same fault plane with respect to the computed paleostress tensor. This parameter determines the robustness of optimization procedure, since faults with $\alpha \leq 30°$ are considered
as compatible with the calculated reduced paleostress tensor.

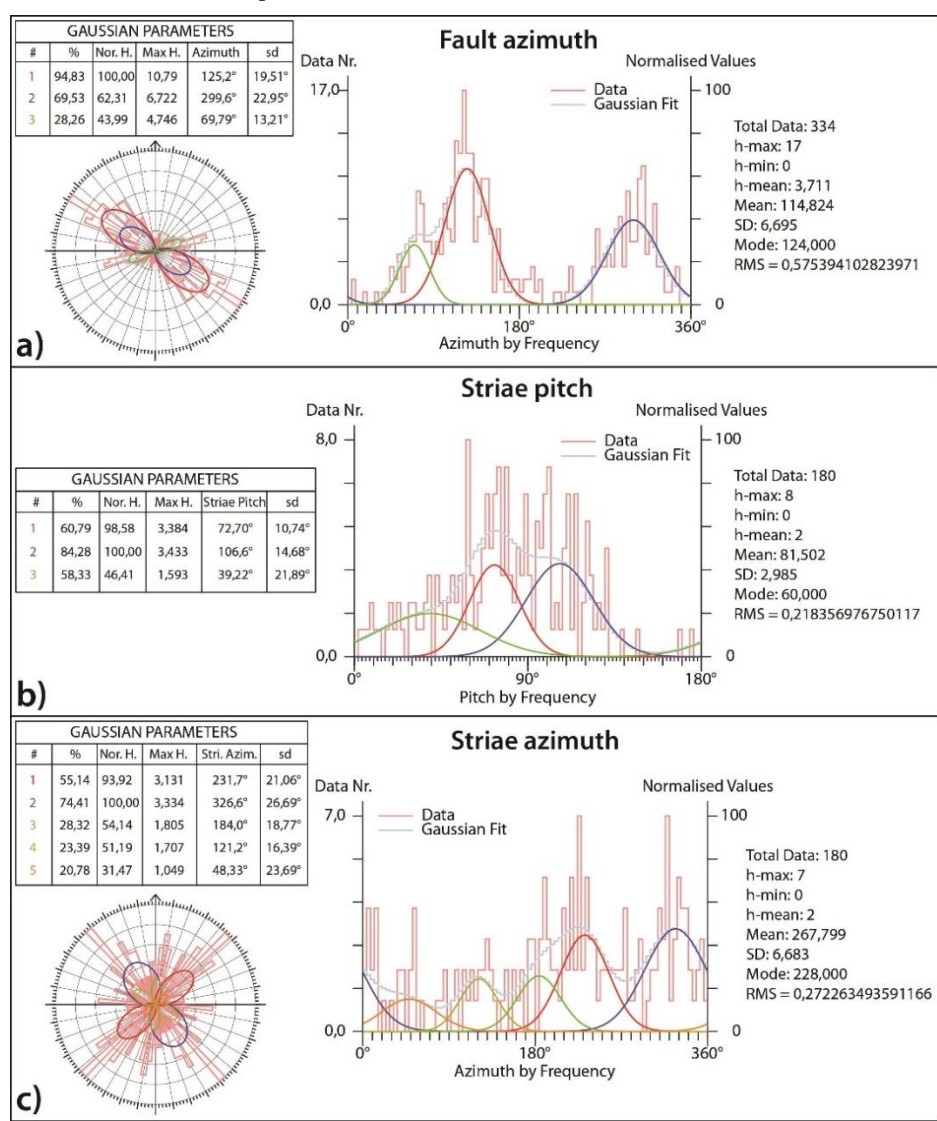

**Figure 5:** Cumulative fault data collected in this study. Polymodal Gaussian distribution statistics of the cumulative fault azimuth (a), slickenside striations pitch (b) and azimuth (c). Statistical analysis was performed using Daisy v.4.1 software (Salvini et al., 1999; Salvini, 2002). The Gaussian Parameters are: number of Gaussian peak (#), percentage of occurrence (%), Normalized Height (Nor.H.), Maximum
Height (Max.H.), and standard deviation (sd).

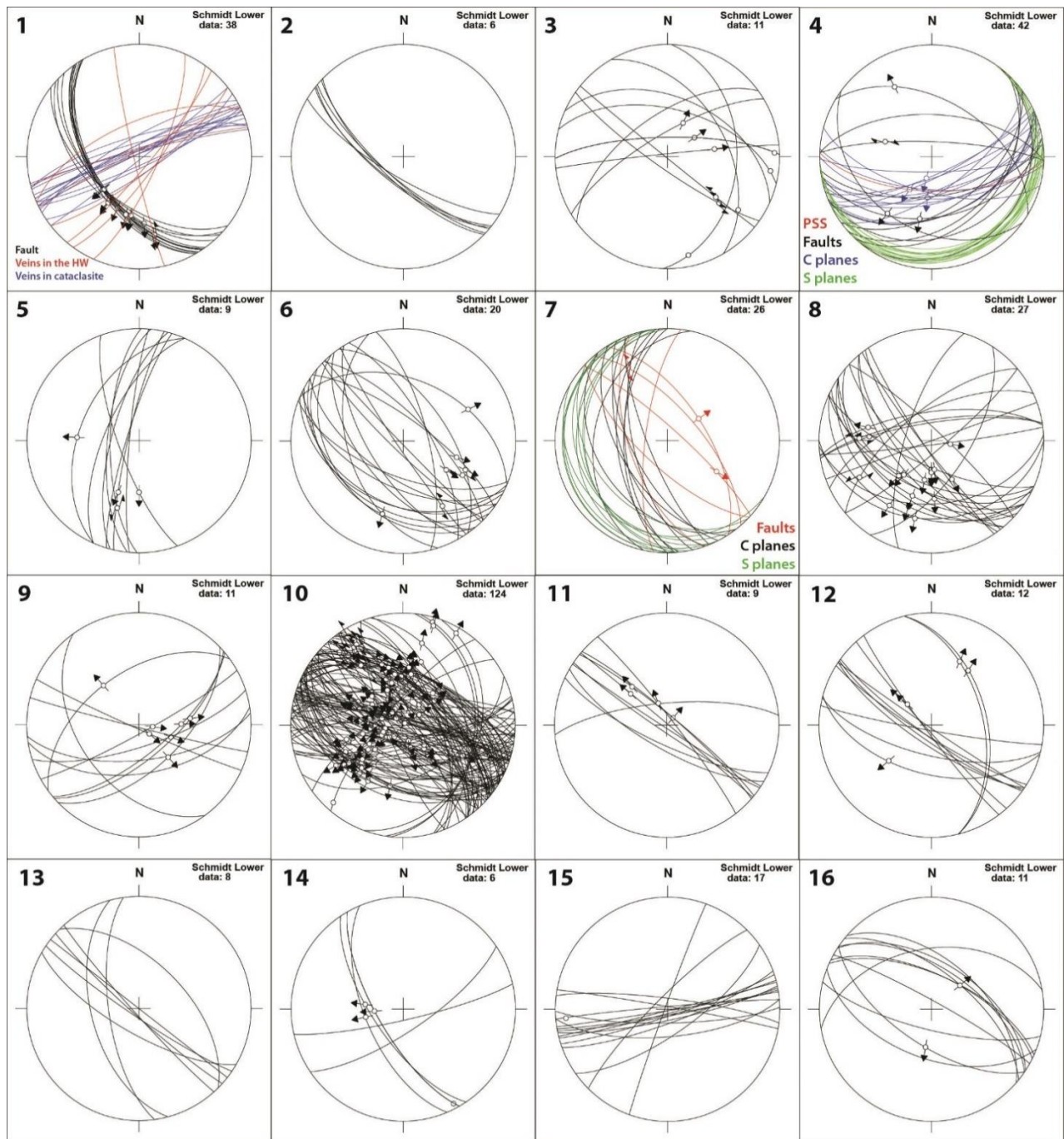

**Figure 6:** Equal area stereoplots (Schmidt net, lower hemisphere) showing the attitude of the main structural features measured in the field and drawn with the Win-Tensor program (Delvaux, 1993). Numbers refer to the sites of structural measurements shown in Fig. 3.


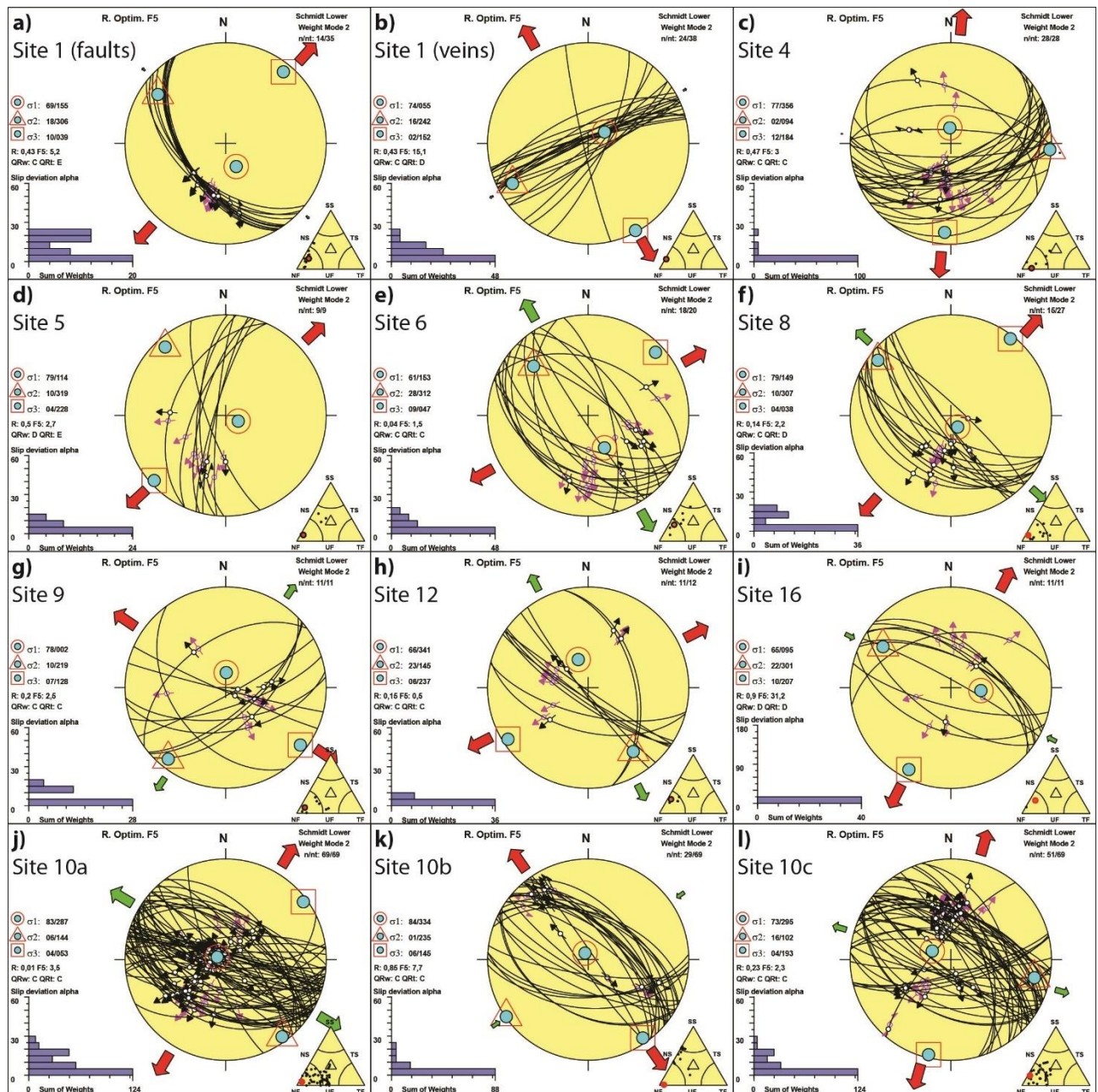

**Figure 7:** Paleostress tensors calculated from the inversion of fault-slip data using the Win-Tensor software (Delvaux, 1993). Stereograms are Schmidt lower hemisphere projections. In the stereograms, black arrows represent the measured hanging wall slip vector on a fault plane; purple arrows represent the theoretical optimal slip vector on a fault plane with respect to the computed paleostress tensor. Site numbers refer to numbers of the structural stations in Fig. 3. In the Frohlich triangles, NF: pure normal faulting; TF: pure reverse faulting; SS: pure strike slip faulting; NS: transtension; TS transpression; UF: radial extension.


## 4. Results

### 4.1. Cumulative fault data analysis

Merging of all fault datasets from the different structural stations allowed us to perform a cumulative statistical analysis on fault data at the scale of the study area (Fig. 5). Distribution of fault azimuth reveals that there are two major trends in fault strike (Fig. 5a). The dominant trend is represented by a vast majority of NW-SE striking faults (N125° and N300°), while a second subsidiary fault population is composed of fault planes striking ENE-WSW (N070°). None of these two trends correlates with the orientation of the morphological boundary between the MMR and the Medio Tiberino and Terni basins

(i.e., ~N-S and ~N100°, respectively).

Three pitch populations have been found for the slickenside striations measured on fault surfaces (73°, 107° and 39°; Fig. 5b). This distribution documents dominant dip-slip over subordinate strike-slip kinematics. Analysis of striae azimuth distribution on normal, oblique/normal and strike-slip faults identified five different populations (N232°, N327°, N184°, N121° and N048°; Fig. 5c). These populations define three main directions of movement along fault planes that are NE-SW (populations 1 and 5

in Fig. 5c), NW-SE (populations 2 and 4 in Fig. 5c) and N-S (population 3 in Fig. 5c).

### 4.2. Fault data and paleostress analysis on single structural stations

In the following, we describe the key structural features observed at selected field stations along the trace of the supposed L-shaped MMFS, starting from its northernmost tip and progressing towards its southeastern end (Fig. 3).

The Viepri site (structural site n. 1 in Fig. 3) is characterized by a SW-dipping normal fault that separates the Jurassic micritic

limestones of the Corniola, in the footwall, from the Eocene marls of the Scaglia Variegata, in the hanging wall (Fig. 8). The principal slip surface (PSS) of the fault is marked by a fine grained cohesive cataclasite formed at the expenses of the Jurassic carbonate (Fig. 8), while above the PSS a few meters thick layer of cataclasite and proto-cataclasite was formed at the expenses of the Eocene marly lithotypes. These latter have a coarser grain size and are less consolidated than those of the PSS and contain an extensional top-to-the-SW S/C fabric (*sensu* Berthé et al., 1979). Slickenside striations on the PSS point to a normal

to normal/oblique sense of shear with a sinistral component (stereonet n. 1 in Fig. 6). Cataclasites in the hanging wall of the PSS and, locally, the PSS itself are cut by cm- to dm-thick NE-SW trending subvertical calcite veins (Fig. 8b). Subvertical calcite veining also occurs in marls of the Scaglia Variegata in the hanging wall of the fault, though with a more diffuse and pervasive spatial distribution (Fig. 8c). The paleostress tensor reconstructed with data from the PSS shows a sub-horizontal NE-SW trending σ3 axis (Fig. 7a), while the opening of tensile veins in the hanging wall of the faults is compatible with a

NW-SE trending σ3 axis (Fig. 7b).

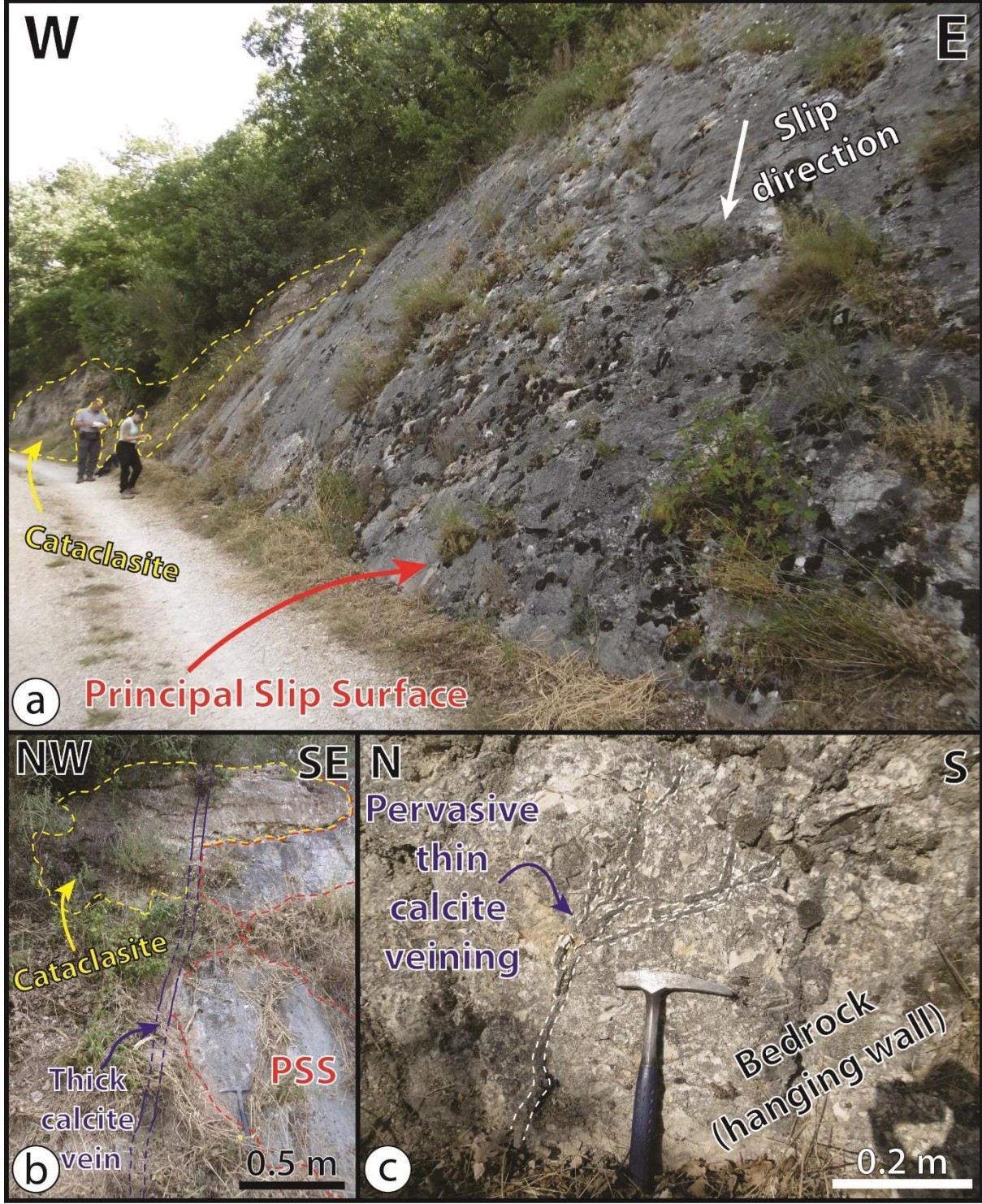

**Figure 8:** Extensional fault surface exposed in the Viepri area (site 1 in Fig. 3) (a) with examples of the calcite veins network hosted in the cataclasite (b) and the hanging wall (c). Structural measurements from this site are reported in stereonet n. 1 in Fig. 6.

East of Massa Martana, on the western slope of the Monte Castro (structural site n. 2 in Fig. 3), mesoscale NW-SE trending normal faults cut through the Lower Jurassic limestone of the Calcare Massiccio (stereonet n. 2 in Fig. 6). Due to the lack of striae on the fault surfaces, the dataset is here too small for a reliable paleostress analysis. However, at the first order the strike of these few faults is generally compatible with a NE-SW trending direction of extension.

In the abandoned quarry to the east of Acquasparta (structural site n. 3 in Fig. 3), the Calcare Massiccio appears to be highly fractured and cut by an intricate tangle of faults, including normal/oblique and subvertical sinistral fault planes (stereonet n. 3 in Fig. 6; Fig. 9). Intense fracturing at the outcrop scale appears to be correlated with diffuse strike-slip and reverse faulting likely related to the orogenic contractional phase. Consequently, it is here hard to distinguish if the oblique/extensional planes are neo-formed or reactivated inherited slip surfaces. At this site, our difficulty in identifying proper fault clusters related to specific tectonic events precluded a reliable paleostress analysis.


**Figure 9:** Examples of syn-orogenic structures in the Calcare Massiccio  in the Acquasparta quarry (site n. 3 in Fig. 3; stereonet n. 3 in Fig. 6). a) Subvertical, NE-SW and NW-SE striking strike-slip faults (marked by red dashed lines), with details of two faults surfaces (in b and c) with strike-slip slickenlines marked by white dashed lines. d) NW-SE trending thrust fault marked by the dashed red line. Note the diffuse brecciation and fracturing at the outcrop scale.

In the Poggio Azzuano area (structural site n. 4 in Fig. 3), at the southernmost tip of the N-S trending segment of the boundary between the MMR and the Medio Tiberino and Terni basins, a S-dipping PSS is found along with extensional top-to-the-S S/C structures within the marly limestone of the Cretaceous Scaglia Rossa  (stereonet n. 4 in Fig. 6; Fig. 10a-c). The paleostress

tensor calculated at this site shows a NNE-SSW direction of extension (Fig. 7c). About 500 m to the SSE of this site, along the road to Cesi (structural site n. 5 in Fig. 2), the computed paleostress tensor is substantially different. The Scaglia Rossa is there cut by N-S normal to normal/oblique faults with a sinistral component (stereonet n. 5 in Fig. 6; Fig. 10d). Paleostress determination from this fault system yielded an extensional regime with a NE-SW trending sub-horizontal $\sigma 3$ axis (Fig. 7d), compatible with that reconstructed from the Viepri site.

In the western part of the Cesi village (structural site n. 6 in Fig. 3), steeply dipping extensional fault planes cut through cohesive cataclasites formed at the expenses of the fine-grained limestone of the Corniola . All these faults strike NW-SE and dip either to the NE with oblique normal/dextral striae, or to the SW with oblique normal/sinistral striae (stereonet n. 6 in Fig. 6). Tensorial analysis performed on this dataset yielded a NE-SW trending sub-horizontal $\sigma 3$ axis, with a steeply dipping $\sigma 1$ axis and a gently dipping $\sigma 2$ axis (Fig. 7e). In this same area (structural site n.7 in Fig. 3), foliated cataclasites in the Corniola

display a penetrative S-C fabric (stereonet n. 7 in Fig. 6; Fig. 10e). This dataset did not allow us to perform a proper tensorial analysis. However, the attitude of the S-C fabric is consistent with an extensional tectonic regime with a NE-SW trending direction of extension.

At the karstic cave of the Grotta Eolia site in the Cesi village (structural site n. 8 in Fig. 3), a complex network of faults belonging to an extensional fault system juxtaposes the Calcare Massiccio in the footwall against cataclastic red marly

limestone in the hanging wall (stereonet n. 8 in Fig. 6; Fig. 10f). Rocks in the hanging wall can be tentatively attributed to the Rosso Ammonitico. Variably oriented high- and low-angle faults inherited from the previous orogenic phase are also present in the cave. NW-SE striking normal to oblique/normal faults, with striae pitch values between ~70° and ~100° (Fig. 10f), are compatible with an extensional paleostress tensor with a NE-SW trending sub-horizontal $\sigma 3$ axis (Fig. 7f), alike the other two above-described sites in the western part of the Cesi village (structural sites n. 6 and 7 in Fig. 6). By contrast, a substantially

different paleostress tensor has been obtained for NE-SW striking extensional faults exposed ~1 km E of Cesi (structural site n. 9 in Fig. 3) and cutting through the micritic limestone of the Maiolica (stereonet n. 9 in Fig. 6). There, paleostress analysis resulted in an Andersonian extensional system with a NW-SE trending direction of extension (Fig. 7g). This paleostress tensor is consistent with that responsible for the formation of the tensile veins at the Viepri site (site n. 1 in Fig. 6 and Fig. 7b).

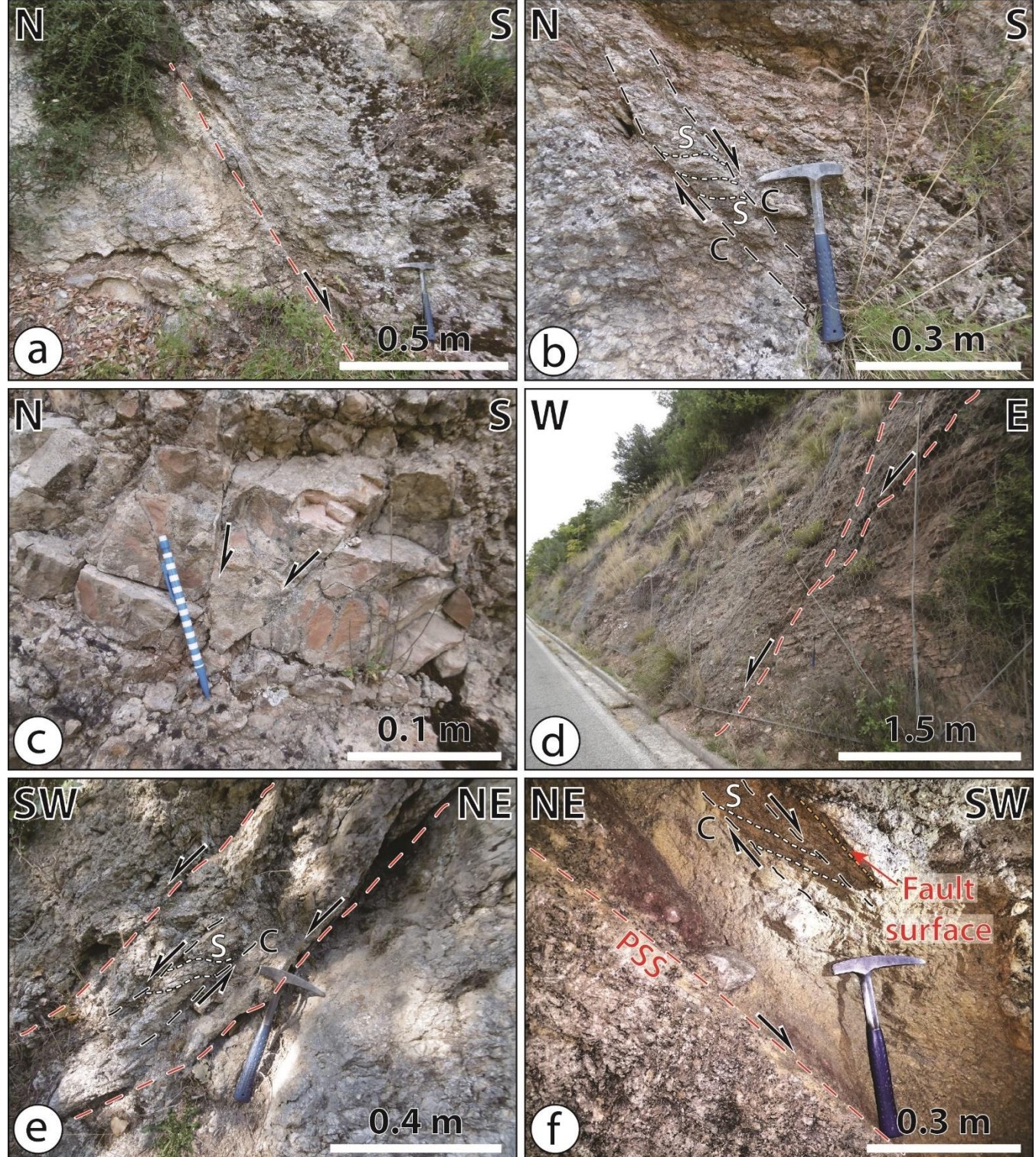

 **Figure 10:** a) S-dipping PSS of a normal fault cutting through the Scaglia Rossa in the Poggio Azzuano area (site n. 4 in Fig. 3; stereonet n. 4 in Fig. 6), with detail of the extensional top-to-the-S S-C fabric developed within the cataclasite (b) and of a second-order conjugate normal faults pair (c) in the hanging wall of the PSS. d) Example of a W-dipping normal fault cutting through the Scaglia Rossa E of Poggio Azzuano (site n. 5 in Fig. 3; stereonet n. 5 in Fig. 6). e) Extensional top-to-the-SW S-C fabric within foliated cataclasite formed at the expenses of the Corniola cropping out W of Cesi (site n. 7 in Fig. 3; stereonet n. 7 in Fig. 6). f) Extensional SW-dipping PSS putting in contact the Calcare Massiccio (in the footwall) with the Rosso Ammonitico (in the hanging wall) in the Grotta Eolia site in Cesi (site n. 8 in Fig. 3; stereonet n. 8 in Fig. 6); note the top-to-the-SW extensional S-C fabric developed within the fine-grained cataclasite of the fault core and a second order SW-dipping fault surface with dip-slip slickenlines (marked by the thin dashed orange line) in the hanging wall of the PSS.

About 1 km to the east of Cesi, the limestone of the Calcare Massiccio exposed in an abandoned quarry (structural site n. 10 in Fig. 3) appears intensely fractured and cut by a dense network of high angle normal, oblique and strike-slip faults (Fig. 11 and stereonet n. 10 in Fig. 6). Some NE-dipping fault planes bear two sets of slickenlines, one related to normal to oblique slip (pitch = 60°-70°) and one related to strike slip (pitch ~5°-25°), with the latter post-dating the first (Fig. 12). We performed tensorial analysis on a dataset of more than one hundred fault planes, most of which strike NW-SE. After data treatment, we distinguished three data subsets, (i) one including normal/oblique fault planes with NE-SW trending slickenlines, (ii) one characterized by sinistral strike-slip faults with NW-SE trending slickenlines, and (iii) one with normal/oblique (with a sinistral component) fault planes with NNE-SSW trending slickenlines (Fig. 7j, k, l, respectively). Tensorial analysis performed on these three data subsets yielded three different andersonian extensional paleostress tensors, with a sub-vertical σ1 axis and a sub-horizontal σ3 axis trending NE-SW, NW-SE, and NNE-SSW respectively (Fig. 7j, k, l). All these results have a satisfying slip deviation angle α < 30°. It is worth noting that these three paleostress tensors reconstructed from this large dataset in a single location are consistent with those calculated in other sites along the MMFS, where datasets are smaller and tensorial analysis only gave one paleostress tensorial solution for each site.

Subvertical NW-SE striking normal/oblique fault planes in the Calcare Massiccio have also been measured close to the Madonna dell'Olivo Sanctuary, in the Piedimonte area (structural site n. 11 in Fig. 3; stereonet n. 11 in Fig. 6). Unfortunately, the dispersion of planes is too small in this dataset to perform a reliable tensorial analysis. At this same location, Brozzetti & Lavecchia (1995) and Bonini et al. (2003) reported mainly E-W to WNW-ESE striking normal/oblique (dextral) faults, compatible with an extensional paleostress tensor with a sub-horizontal NE-SW σ3 axis (Brozzetti & Lavecchia, 1995).

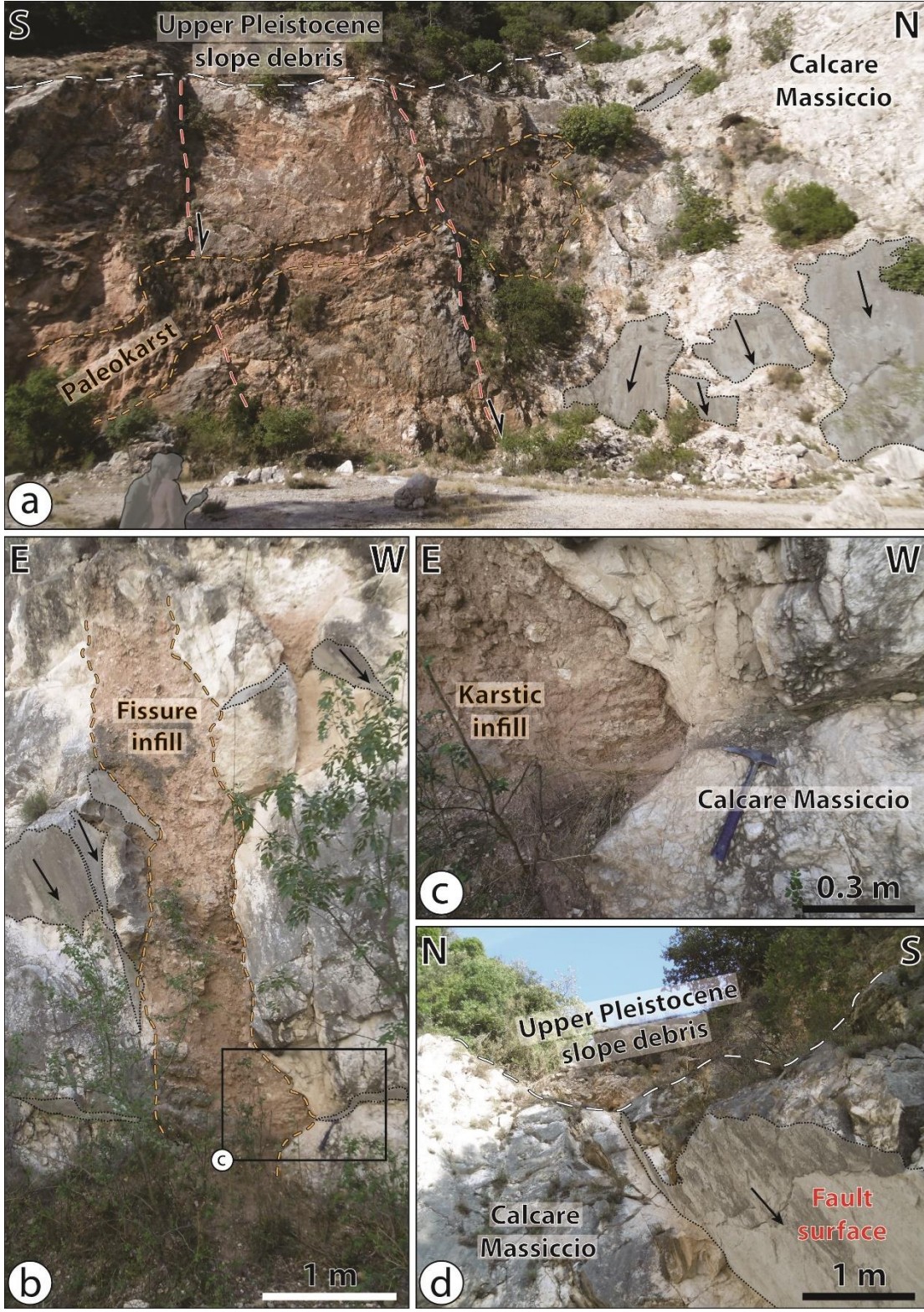

**Figure 11:** Outcrop pictures from the Cesi quarry (site 10 in Fig. 3). Structural measurements from this site are reported in stereonets n. 10 in Fig. 6. Red arrows indicate extensional fault planes in the Calcare Massiccio. a) Panoramic view of the western wall of the quarry, with a south-dipping paleo-karstic network with speleothems developed in the faulted limestone of the Calcare Massiccio; note that Upper Pleistocene slope debris suture normal fault planes. b) Example of a sub-vertical, non-faulted, Quaternary fissure infill within the Calcare Massiccio. c) Detail of the contact between a fissure infill and the Calcare Massiccio (see Fig. 5b for location); note that the faults and

fractures affecting the limestone do not propagate within the karstic infill of the fissure. d) Example of Upper Pleistocene slope debris suturing a normal fault cutting through the Calcare Massiccio. Shaded gray polygons show fault surfaces with black arrows showing the slip direction of the hanging wall block.

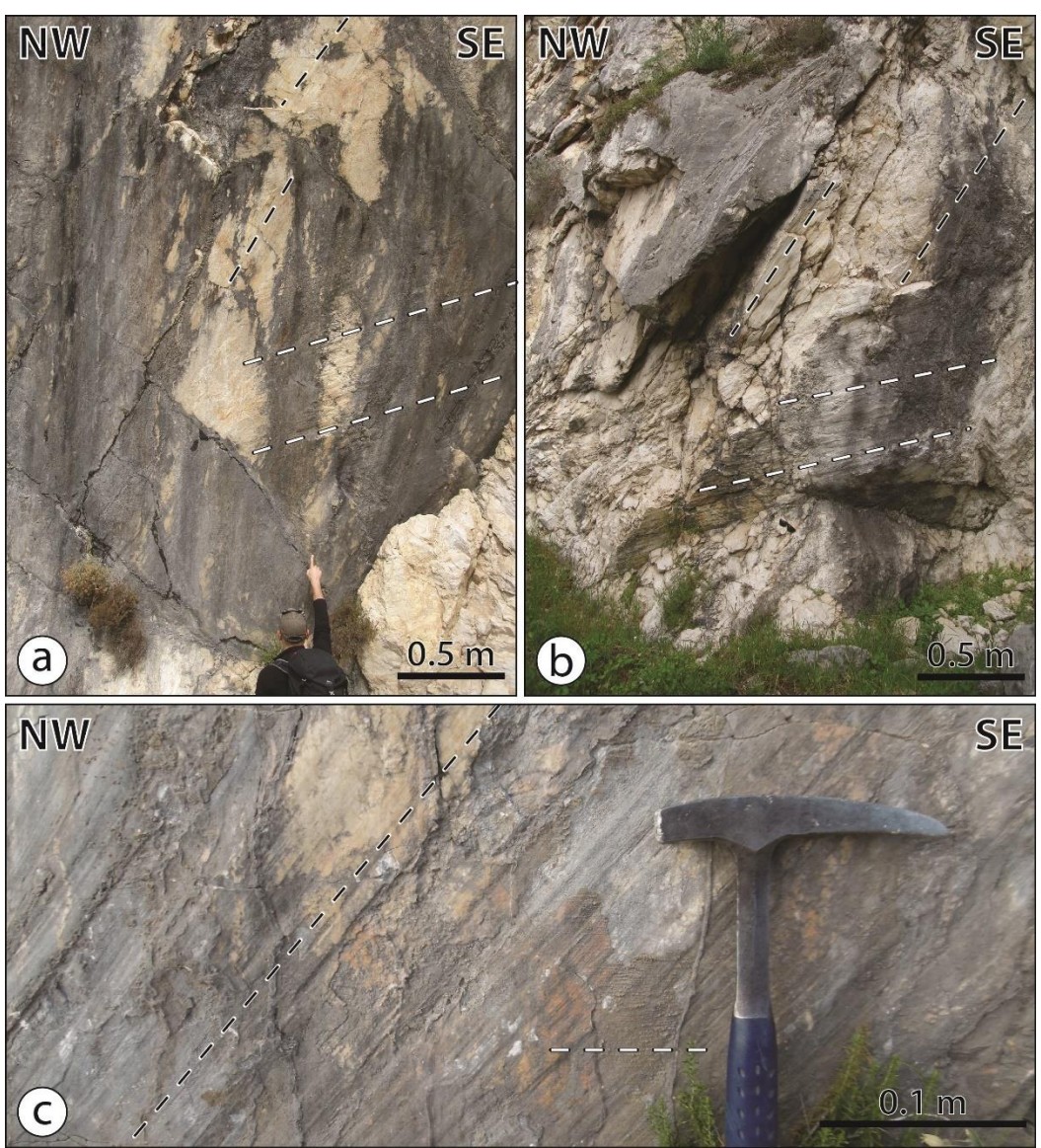

**Figure 12:** Examples of steeply-dipping, NW-SE striking faults in the Cesi quarry (site n. 10 in Fig. 3; stereonet n. 10 in Fig. 6) with a later generation of strike-slip slickenlines (marked by white dashed lines) overprinting an older generation of normal-to-oblique-slip slickenlines**.**

At structural site n. 12 (Fig. 3), NW-SE striking extensional fault planes cut through the limestone of Calcare Massiccio

(stereonet n. 12 in Fig. 6). The paleostress tensor reconstructed at this site has a NE-SW trending sub-horizontal σ3 axis (Fig. 7h), even though the σ1 axis is not exactly subvertical (dip angle 66°). An analogue set of faults has been measured at structural site n. 13 (Fig. 3; stereonet n. 13 in Fig. 6). These extensional faults cut through the brecciated Corniola and are sutured by Upper Pleistocene slope debris (Fig. 13). The brecciated Corniola consists of a monomictic, angular and heterometric breccia not exceeding 2-3 m in thickness. The clasts are derived from the underlying bedrock and exhibit a chaotic fabric, suggesting

that the breccia formed locally (in situ), although a limited remobilization of clasts cannot be entirely ruled out. Unfortunately, no slickenlines were observed at this site, thus preventing tensorial analysis.

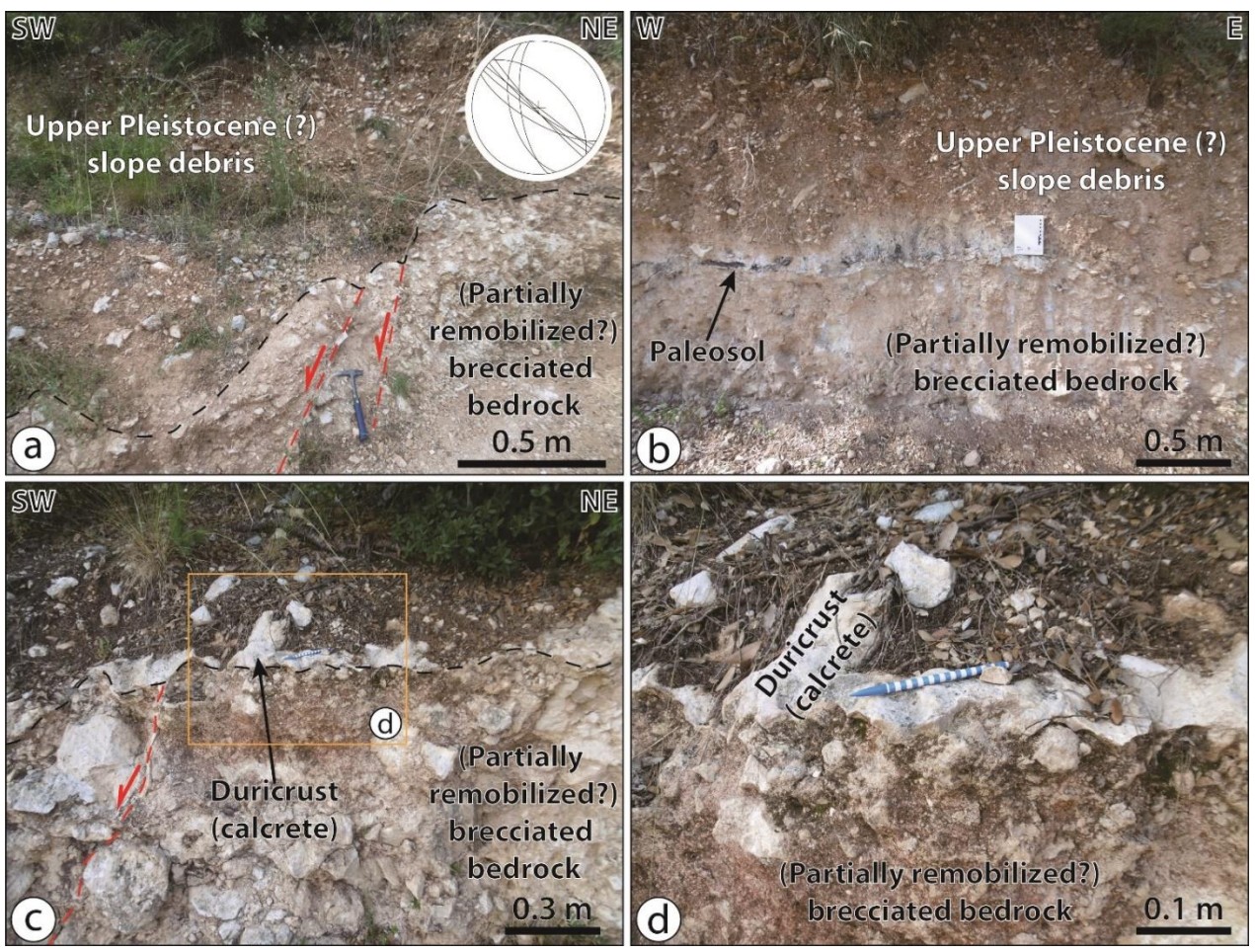

**Figure 13:** Outcrops in the Fontana della Madonna area (site n. 13 in Fig. 3). a) Example of extensional faults cutting through the

Mesozoic carbonate substratum and being sutured by Upper Pleistocene (?) slope debris, with stereographic projection (Schmidt net, lower hemisphere) of fault data from this site. b) Dark brown paleosol at the contact between the brecciated Mesozoic bedrock and the overlying Upper Pleistocene slope debris. c-d) Carbonate duricrust (calcrete) locally hardening the paleosol between the brecciated Mesozoic bedrock and the overlying Upper Pleistocene slope debris and suturing normal faults cutting through the carbonate substratum.

A similar situation has been observed in the easternmost structural site at the eastern tip of the MMFS(Rocca San Zenone area, structural site n. 14 in Fig. 3), where extensional, oblique, and strike-slip faults cut through the brecciated Corniola and are sutured by Upper Pleistocene slope debris (stereonet n. 14 in Fig. 6). On a single NW-SE striking and SW dipping fault plane, three different generations of striae have been observed, pointing to a polyphase reactivation of this fault plane with a normal, oblique (normal/dextral) and strike-slip movement (Fig. 14). However, the dispersion of planes in this dataset is too small to

generate a reliable tensorial analysis.

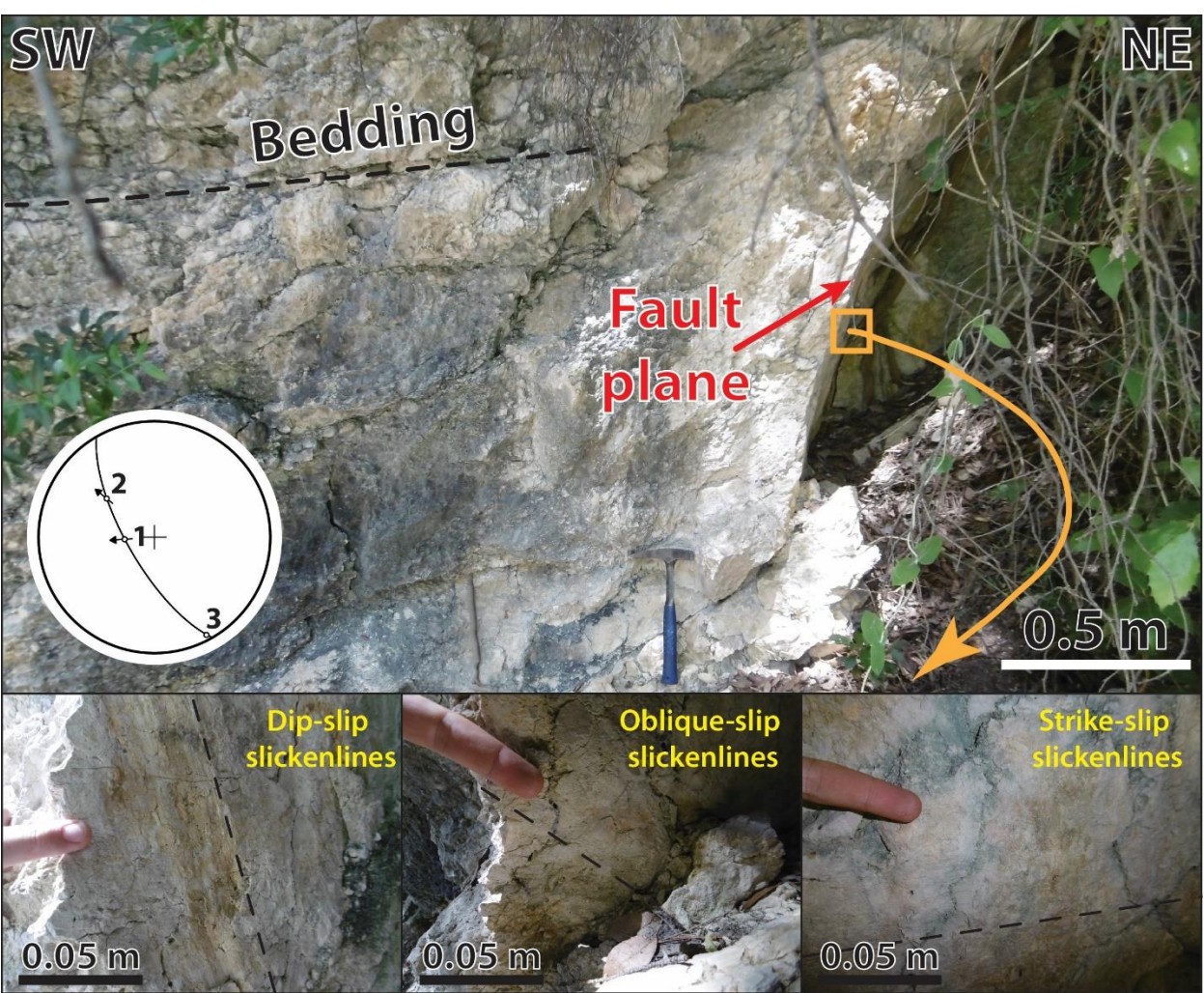

**Figure 14:** Example of a fault plane in the Trevi area cutting through the Mesozoic carbonate substratum and displaying three different sets of slickenlines (site 14 in Fig. 3). Structural measurements from this site are reported in stereonet n. 14 in Fig. 6.


As to extensional deformation in the Plio-Quaternary fill of the Medio Tiberino and Terni basins, brittle structures were only observed in the San Gemini area cutting through Upper Pliocene/Lower Pleistocene yellow sandy conglomerates, likely belonging to the Fosso Bianco Fm *sensu* Basilici (1992, 1997) (Fig. 15). These structures consist of subvertical fractures within the San Gemini area (structural site n. 15 in Fig. 3; stereonet n. 15 in Fig. 6) and normal faults exposed in an abandoned quarry south of San Gemini (structural site n. 16 in Fig. 3; stereonet n. 16 in Fig. 6). Tensorial analysis performed on this latter fault set provided an extensional paleostress tensor with a sub-horizontal NE-SW trending σ3 axis (Fig. 7i).

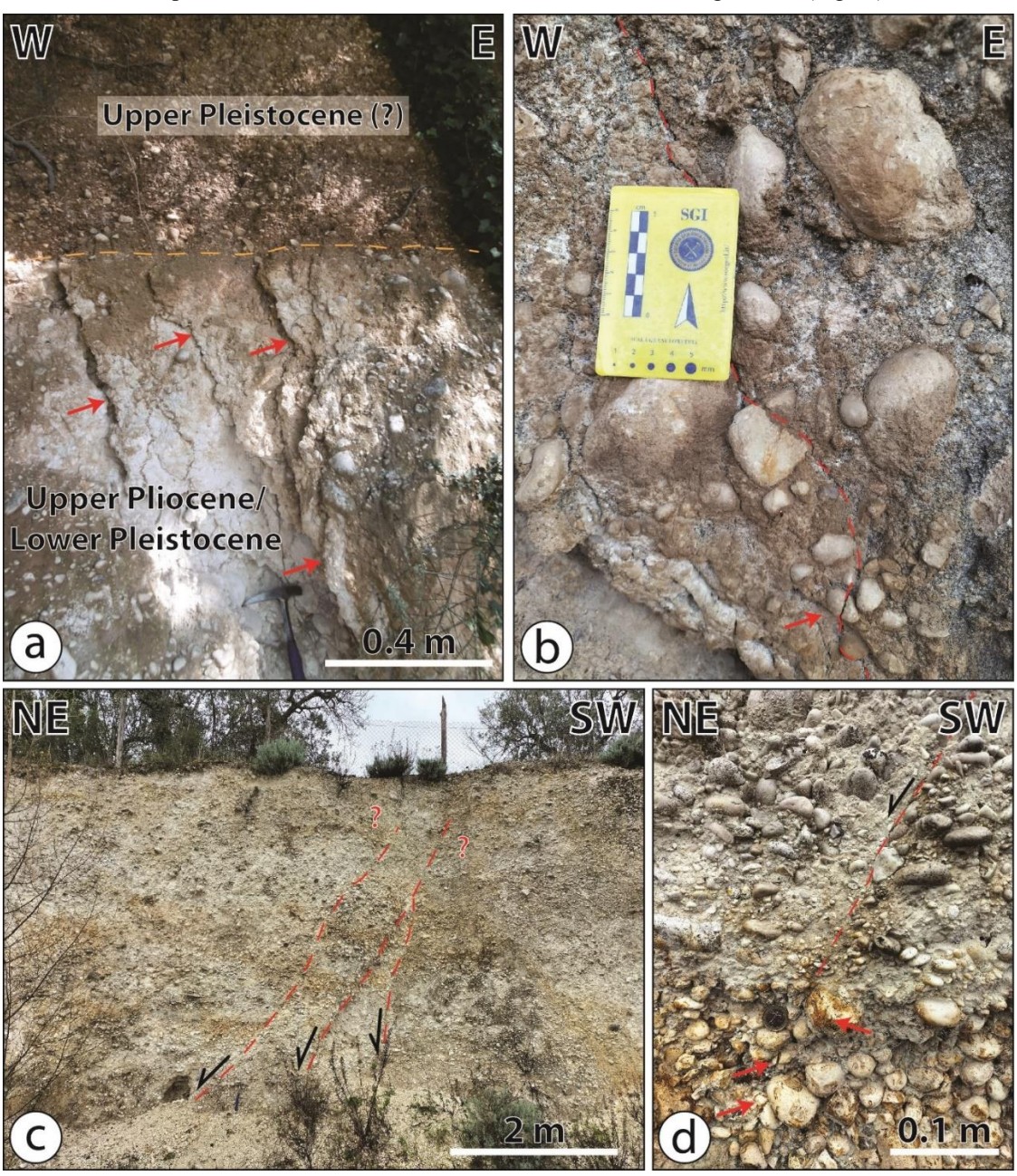

**Figure 15:** Examples of brittle structures within Upper Pliocene/Lower Pleistocene yellowish conglomerates in the San Gemini area. a) Sub-vertical fractures (indicated by red arrows) within Upper Pliocene/Lower Pleistocene conglomerates sutured by Upper Pleistocene (?) reddish alluvial conglomerates in San Gemini (site 15 in Fig. 3; stereonet n. 15 in Fig. 6). b) Detail of an irregular sub-vertical fracture within Upper Pliocene/Lower Pleistocene yellowish conglomerates at the same site as (a); the red arrow points to a broken pebble straddling the fracture plane. c) NE-dipping normal faults within Upper Pliocene/Lower Pleistocene conglomerates exposed in an abandoned quarry south of San Gemini (site 16 in Fig. 3; stereonet n. 16 in Fig. 6); note that upward fault throw appears progressively less evident. d) detail of a NE dipping normal fault in the same site as (c); note the broken pebbles marked by red arrows in the downward continuation of the fault plane (marked by a dashed read line).

## 4.3. Relationships between extensional faults and the Plio-Quaternary sediments

In this study, for the Plio-Quaternary age of the continental sedimentary units of the Medio Tiberino and Terni basins we have adopted the stratigraphic schemes of Basilici (1997), Bonini et al. (2003) and the 1:10.000 scale geological map of the Regione Umbria – Servizio Geologico (2013). All these studies agree on the age of the deposits described in the San Gemini area and along the western and southern margins of the MMR. The majority of the studied fault surfaces along the MMFS cut through the Meso-Cenozoic carbonate sequences of the MMR. Very few faults have instead been documented in the Plio-Quaternary infill of the Medio Tiberino and Terni basins, and, in no case, have faults been observed cutting through deposits younger than the Upper Pliocene/Lower Pleistocene. In the rare cases where the relationships between continental deposits and fault planes can be clearly observed, the former appear to suture the faults. This is clearly visible at two sites located along the eastern part of the ESE trending segment of the slopes of the MMR in the Fontana della Madonna and Rocca San Zenone areas (sites 13 and 14 in Fig. 3). There, Upper Pleistocene slope debris deposits suture NW-SE striking normal to oblique faults cutting limestones of the Meso-Cenozoic substratum (Fig. 13a). In particular, in the Fontana della Madonna area (site n. 13 in Fig. 3), a dark brown paleosol developed at the transition between the brecciated Mesozoic carbonate bedrock and the overlying Upper Pleistocene (?) slope debris (Fig. 13b). This paleosol is locally capped by a carbonate duricrust (calcrete) that sutures SW-dipping normal faults cutting through the brecciated Mesozoic carbonate substratum (Fig. 13c-d). Moreover, in an abandoned quarry to the east of Cesi, a paleo-karstic network is exposed (Fig. 11a). The orientation of the paleo-karstic system seems structurally controlled by NW-SE trending normal to oblique faults. However, we observed neither deformation on the exposed speleothems or in the karst-filling sediments (Fig. 11b, c) nor displacement within the overlying poorly consolidated, Upper Pleistocene reddish slope debris (Fig. 11d). No deformation has been observed affecting the speleothems of the Grotta Eolia either (structural site n. 8 in Fig. 3).

Although extensional fault systems cutting through the oldest deposits of the Plio-Quaternary sedimentary infill and compatible with a NE-SW direction of extension are reported in the literature (Brozzetti & Lavecchia, 1995; Brozzetti & Stoppa, 1995; Basilici, 1997), very little evidence of deformation has been found in the continental deposits of the Medio Tiberino and Terni basins, mainly due to the general lack of informative outcrops. Most of the sites reported in the literature are actually located in ancient or still active quarries. The only clear field evidence of Quaternary deformation is in the southern part of San Gemini (structural sites 15 and 16 in Fig. 3). There, E-W trending subvertical joints cut through Upper Pliocene/Lower Pleistocene

well consolidated yellowish conglomerates (Fig. 15a-b; stereonet n. 15 in Fig. 6). These discontinuities are in turn sutured by Upper Pleistocene (?) poorly consolidated reddish alluvial conglomerates (Fig. 15a). Moreover, in an abandoned quarry right

to the south of San Gemini, NW-SE striking normal faults cut through the Upper Pliocene-Lower Pleistocene conglomerate (stereonet n. 16 in Fig. 6; Fig. 15c-d). Some of these faults are characterized by a progressive upward reduction of the vertical throw, thus suggesting a syn-sedimentary fault activity (Fig. 15c).

## 5. Discussion

### 5.1. Discrepancy between the morphostructural trends and the strike of single fault segments

After compiling the structural data along the trace of the MMFS, a striking observation is that there is no evident correlation between the morphostructural trend of the western and southern margins of the MMR and the strike of the measured meso-scale structures. In fact, in only very few cases the strike of the measured extensional faults correlates with the ~N-S or WNW-ESE trends of the main inferred segments of the MMFS. The dominant measured fault strike is NW-SE (Fig. 5a). Moreover, ~E-W striking faults exist along the ~N-S segment of the fault system (as in the case of the Poggio Azzuano area; structural

site n.4 in Fig. 3). Also, ~N-S striking faults occur along the WNW-ESE segment (as in the case of the Cesi or Rocca San Zenone areas; structural sites n.8, 12 and 13 in Fig. 3). This suggests, therefore, that the morphostructural trend of the western and southern margins of the MMR does not correspond to continuous and several kilometer-long fault traces aligned along the ~N-S or WNW-ESE directions. Rather, the MMFS appears to be formed of several disconnected fault segments with different orientations, most of which are aligned with the main structural trend of the Plio-Quaternary extensional structures of the

Northern and Central Apennines (i.e. NW-SE; e.g., Galadini & Galli 2000; Barchi, 2010).

Thus, our results show that the contact between the Upper Pleistocene-Holocene deposits of the Medio Tiberino and Terni basins and the Meso-Cenozoic carbonate and siliciclastic rocks of the MMR is an unconformable stratigraphic contact and not an extensional syn-sedimentary tectonic contact as proposed by earlier studies (e.g., Ambrosetti et al., 1987b; Bonini et al., 2003). In other words, we propose that the MMFS somehow steered the deposition of (at least part of) the Plio-Quaternary

infill of the Medio Tiberino and Terni basins, although this does not imply that this tectonic control was exerted by one single major fault bounding the MMR.

In addition to the geometric misfit between the morphostructure at the large scale and the faults observed in the field as per discussion above, also the distribution of the epicenters of the major historical and instrumental earthquakes from the area corroborates this apparent misfit, as they seem to not align along the morphostructural boundary between the MMR and the

Medio Tiberino and Terni basins (ISIDe Working Group, 2007; Rovida et al., 2020, 2022). At least two clusters of epicenters can be identified east of Massa Martana, in the inner part of the MMR, and in the Terni area (Fig. 2). Moreover, the focal mechanisms of the 2006 San Gemini and 2014 Acquasparta earthquakes yield fault plane solutions that are misoriented to the main structural trend of the western boundary of the MMR (Scognamiglio et al., 2006; Fig. 2). In any case, the clustering of

the instrumental epicenters in the study area is clearly not coherent with the supposed L-shaped geometry commonly attributed in the literature to the MMFS.

## 5.2. What does control the orientation of faults in the Monti Martani Ridge? The role of structural inheritance

If not governed by the active seismotectonics, the orientation of the main structural trends affecting the MMR and its L-shaped geometry may instead be possibly considered as the long-lived expression of the inherited pre-orogenic structural grain. In particular, we refer to the regional scale, ~E-W and ~N-S trending tectonic lineaments that are thought to have controlled the Jurassic paleogeography in the Monti Martani, Sabina and Monti Reatini area, after the dismembering of the Calcare Massiccio carbonate platform during the Early Jurassic rifting phase (e.g., Coltorti & Bosellini, 1980; Calamita et al., 1991; Cipriani et al., 2020). These structural trends match those of the Jurassic extensional faults described within the MMR (Bruni et al., 1995; Coltorti et al., 1995). Several studies have suggested that such structural inheritance may have influenced the nucleation and geometry of the main orogenic fronts during the Apennines nappe-stacking phase (e.g., Decandia & Tavarnelli, 1991; Tavarnelli, 1996; Butler et al., 2006; Scisciani, 2009, Scisciani et al., 2014; Tavarnelli et al., 2019; Curzi et al., 2024), including in the Monti Martani area (Calamita & Pierantoni, 1994, 1995; Bruni et al, 1995; Bonini, 1998). Remarkably, these studies have not reported any significant changes in the orientation of the pre-orogenic features that could be attributed to the syn-orogenic phase.

In light of these considerations, we suggest that, in this area, the pre-orogenic Jurassic tectonic grain might have exerted a fundamental structural influence not only on the Apenninic convergent phase, but also on the Plio-Quaternary post-orogenic extensional phase, as suggested by recent studies in other portions of the Apennines belt (e.g. Tavani et al., 2021; Mercuri et al., 2024). This would agree with previous studies suggesting that the area of the Plio-Quaternary Terni Basin represented an early ~E-W trending structural low between the Monti Martani and the Sabina Plateau pelagic carbonate platforms already in the Jurassic (Cipriani et al., 2020). This depression may have been reactivated during the post-orogenic phase.

Numerical and analogue models of oblique rifting have shown that, in the presence of inherited structural features oriented at a substantial obliquity to the regional extension direction (i.e. ~45°), the bulk large-scale morphostructural trend of the rift system is indeed governed by the orientation of the inherited structural grain, rather than by the regional tectonic trends (e.g. McClay & White, 1995; Keep & McClay, 1997; Brune, 2014). These studies have also shown that the strike of the neo-formed normal faults tends to be almost orthogonal to the regional $\sigma 3$ direction (~15° of deviation; see Brune, 2014). However, analogue models have also documented that, in the case of multiphase extension with different stretching direction, the impact of the inherited structures on the strike of the newly formed faults depends on the amount of extension during the first rifting phase (Wang et al., 2021). Numerical models have suggested that, in such cases, in the absence of variations in the orientation of the regional stress field during rifting, fault kinematics may evolve from a first stage dominated by normal faulting to a second stage where normal, oblique/normal and strike-slip faulting coexist and the strike of the faults may vary from almost orthogonal to the regional $\sigma_3$ direction to broadly parallel to the direction of the inherited structural features (Brune, 2014,

2016). This implies that the orientation of locally derived paleostress tensors computed from limited fault-slip data may deviate during rift evolution from the first-order regional stress field that determined the overall tectonic regime.

The setting described by these models is comparable to the context of the MMFS in the general tectonic framework of the transition between the Northern and Central Apennines. In fact, as discussed above, the MMR is characterized by significant

structural inheritance due to the Jurassic rifting-related ~E-W and ~N-S striking faults. These structures have a moderate obliquity to the Quaternary and present-day regional $\sigma_3$ direction related to post-orogenic extension in the internal part of the orogen that is NE-SW (e.g., Piccardi et al., 1997; Montone et al., 2012; Mariucci & Montone, 2024 and references therein). We thus suggest that the observed complex structural pattern of the MMFS might result from the interaction between inherited pre-orogenic structural features and post-orogenic regional stress tensor oriented obliquely to the pre-to-syn-orogenic structural

grain.

Mercuri et al. (2024) documented the reactivation of Jurassic normal faults belonging to a ~10 km long fault system (Celano-Ovindoli-Pezza Fault System) in an internal portion of the Central Apennines again as extensional faults during post-orogenic extension. In that case, the strike of the inherited normal fault(s) (i.e., NNW-SSE) is very little oblique to the ideal strike of normal faults formed as optimally oriented in the framework of the post-orogenic extensional stress field, that is, NW-SE. This

might have mechanically favored reactivation of pre-existing structures over the formation of neo-formed optimally oriented faults. By contrast, in the case of the MMR, the strike of the inherited Jurassic faults (i.e., ~N-S and ~E-W; Bruni et al., 1995; Coltorti et al., 1995) exhibits moderate- to high obliquity to the NW-SE direction, which would be the optimal direction for neo-formed normal faults in the present-day extensional stress field. Thus, this might have favored neoformation of optimally oriented faults over reactivation of pre-existing misoriented faults (Fig. 16). This would agree with the observation that most

of the faults documented in this study strike NW-SE (Fig. 5a) and, accordingly, most of them have dip-slip slickenlines with oblique to strike-slip striae being clearly subordinate in our dataset (Fig. 5b). Also, the presence of a non-optimally oriented inherited structural grain seems to have favored the formation of a very segmented fault system, composed of a patchwork of relatively short (a few kilometers long?), optimally oriented normal faults, rather than the formation of long extensional faults such as those known in other parts of the internal domain of the Apennines (Fig. 16). This interpretation might have relevant

implications on seismic hazard assessments, since the length of seismogenic sources correlates linearly to the maximum potential magnitude of earthquakes (e.g., Wells & Coppersmith, 1994).

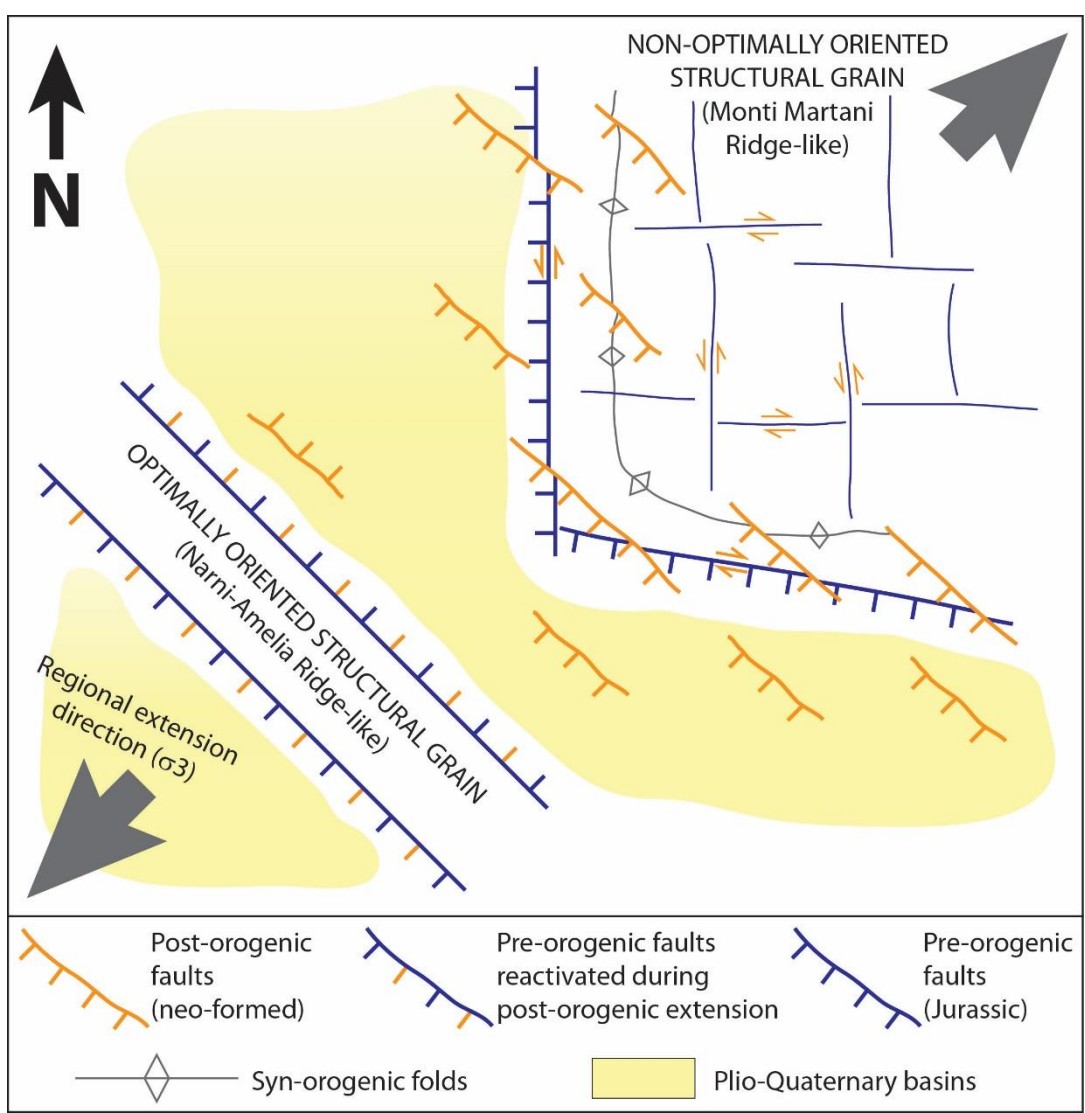

**Figure 16:** conceptual model for the development of post-orogenic extensional faults in the Monti Martani area. Optimally oriented structures are preferentially reactivated during post-orogenic extension and favor the formation of longer seismogenic normal faults. By contrast, a non-optimally oriented inherited structural grain with respect to the regional extensional stress field favors the activation of shorter, neo-formed normal faults which are ca. perpendicular to the regional extension direction.

## 5.3. Paleostress evolution

Paleostress analyses on the acquired fault-slip data revealed three distinct reduced extensional tensors, with a sub-horizontal $\sigma 3$ oriented NE-SW, NNE-SSW or NW-SE (Figs. 7 and 17). Despite field observations did not allow us to establish the relative chronology between the two first extensional phases, the latter (i.e., the NW-SE $\sigma_3$) clearly postdates the first two. In fact, the extensional paleostress tensor with a sub-horizontal $\sigma 3$ oriented NW-SE has been reconstructed in the Cesi quarry based on

sub-horizontal slickenlines that postdate normal to oblique/normal slickenlines on NW-SE striking polyphase fault surfaces.

The same relative chronology between normal/oblique and sinistral strike-slip striae on ~N-S striking faults has been also previously reported form other sites along the Martani Fault System (Bonini et al., 2003). Moreover, in the Viepri area sub-vertical calcite veins formed under NW-SE trending extension clearly postdate the formation of cataclasite related to normal faults formed under a NE-SW extensional regime (Fig. 8). Thus, compared to previous structural studies on the MMFS (Fig. 17), our work allowed us to better define the evolution of the local paleostress tensor as belonging to a transition from orogen-

orthogonal extension (from NE-SW to NNE-SSW directions of extension) to a previously unknown local phase of orogen-parallel extension (NW-SE direction of extension).

The paleostress tensors with a sub-horizontal NE-SW and NNE-SSW $\sigma 3$ do not necessarily relate to two distinct tectonic phases. In fact, as discussed above, numerical models have shown that in the case of oblique rifting, with the regional extension direction oblique (~45°) to inherited structures, the orientation of the local extensional stress tensor may evolve through time

and pass from slightly oblique to the regional extension direction, to almost perpendicular to the inherited structures (Brune, 2014, 2016). Thus, both local paleostress tensors might relate to the same phase of extension with a regional sub-horizontal minimum stress axis oriented NE-SW and compatible with the overall post-orogenic tectonic regime of the internal part of the Northern and Central Apennines (Mariucci et al., 2024).

Irrespective of whether this interpretation is correct, this mechanism cannot be used to explain the extensional paleostress

tensor with a sub-horizontal $\sigma 3$ oriented NW-SE. This paleostress tensor refers to an extensional tectonic event that postdates the formation of the main NW-SE striking faults that developed under overall NE-SW extension. Moreover, all the measured faults are sutured by Upper Pleistocene deposits (Figs. 11, 13 and 15) and no fault has been observed to dislocate this latter sedimentary unit. Thus, the paleostress tensor constraining NW-SE extension is likely related to a short-lived tectonic pulse of orogen-parallel extension that took place at some point between the Early and the Late Pleistocene. The geodynamic causes

of this orogen-parallel extensional event remain unclear. However, this event may possibly correlate chronologically with a regional phase of increased uplift rate and increased normal fault activity affecting the Northern Apennines during the Early–Middle Pleistocene (Dramis, 1992).

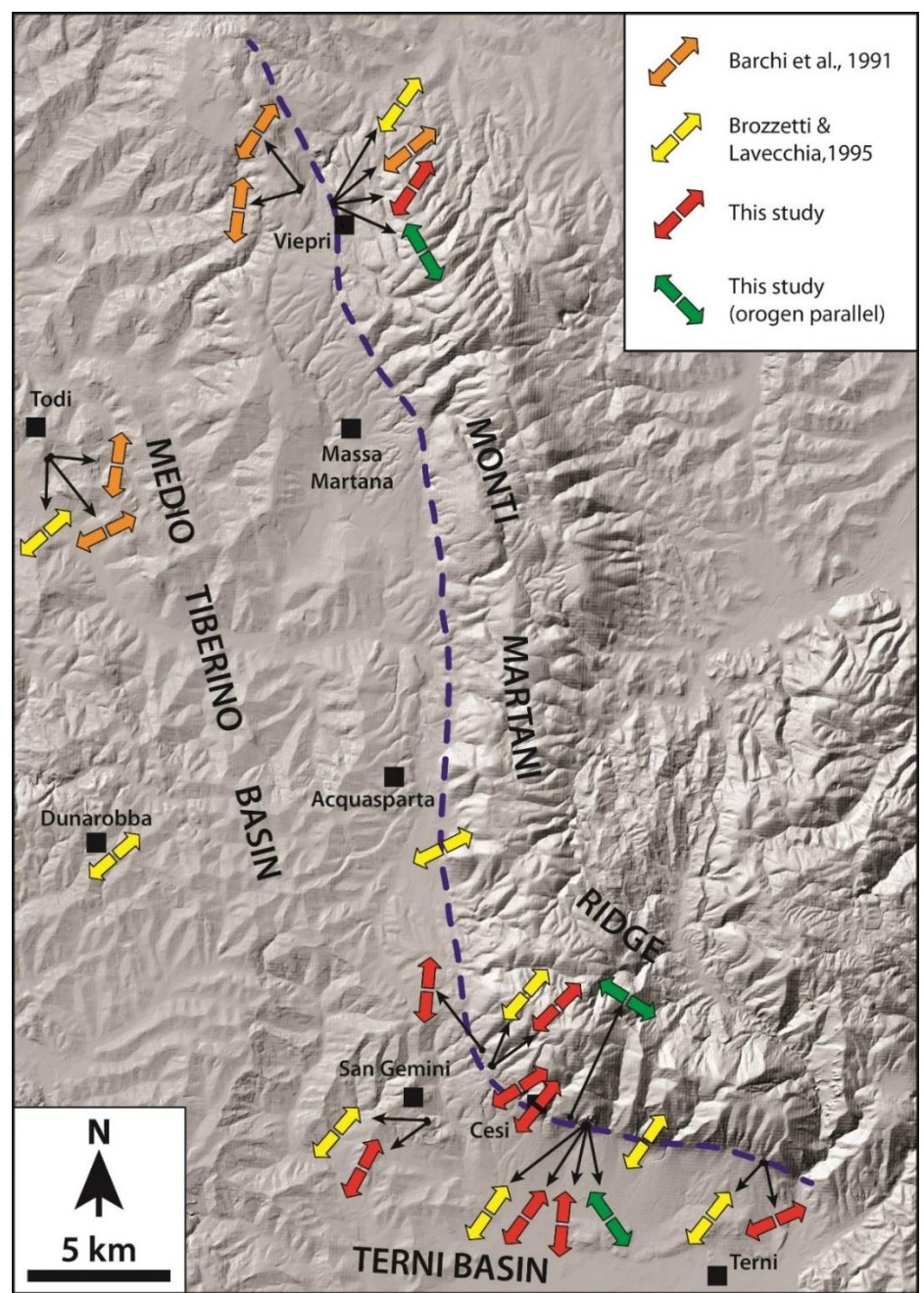

**Figure 17:** Extension directions calculated after paleostress analyses of Plio-Quaternary faults of the Martani Fault System represented on a DTM of the study area. Orange arrows are from Barchi et al. (1991); yellow arrows are from Brozzetti & Lavecchia (1995); red arrows are orogen-orthogonal directions from this study; green arrows are orogen-parallel directions from this study. Dashed blue line marks the morphological boundary between the Monti Martani Ridge and the Medio Tiberino and Terni Plio-Quaternary basins.

The co-existence of orogen-parallel and orogen-orthogonal extensional structures has been already documented in several parts of the Southern Apennines. There, some authors proposed that orogen-parallel extension preceded orogen-orthogonal extension and developed in response to the progressive thrust-belt bending after the opening of the Southern Tyrrhenian Sea during Early Miocene – Pliocene times (Oldow et al., 1993; Ferranti et al., 1996; Ferranti & Oldow, 1999). A similar interpretation cannot be applied to the orogen-parallel extensional event of our study, because oroclinal bending in the internal

Umbria-Marche portion of the Northern Apennines that caused arcuation of the orogen occurred during the Late Miocene – Pliocene (Caricchi et al., 2014), and is thus older than the timing we frame the orogen-parallel extensional event in the MMR documented in this study. However, studies on recent seismic sequences along the N-S trending Ortona-Roccamonfina tectonic line, at the boundary between the Central and Southern Apennines (Fig. 1a), have shown that orogen-parallel and orogen orthogonal extension can locally coexist in an overall tectonic context of orogen-orthogonal extension (Milano et al., 2002,

2008). This geological context has much affinity with the tectonic setting of the Monti Martani area in terms of orientation of the post-orogenic extensional stress tensor and of the structural inheritance. In fact, they document seismic sequences in the area of the Ortona-Roccamonfina lineament, which represents a structural inheritance oriented N-S and subject to a NE-SW oriented post-orogenic extensional stress field. Similarly, the area of the Monti Martani is located in the vicinity of the Olevano-Antrodoco lineament, which represents the boundary between the Northern and Central Apennines (Parotto & Praturlon, 1975;

Castellarin et al., 1978; Cosentino et al., 2010; Fig. 1a) and also represents a strong N-S oriented structural inheritance and subject to a NE-SW oriented post-orogenic extensional stress field.

Another possible explanation for the orogen-parallel extensional event documented in this study is that it represents local perturbations of the regional stress field in relay zones between extensional faults in a mature stage of their evolution. In fact, in classical examples of release faults in extensional settings (e.g., Destro, 1995; Roberts, 1996; Destro et al., 2003), fault slip

directions measured close to the main faults tips or along cross faults can differ by almost 90° from the regional direction of extension. Also numerical models have shown that such perturbations of the regional stress field preferentially occur at the tips of isolated faults, or in relay zones between overlapping faults (Kattenhorn et al., 2000). This mechanism can also account for the coexistence of orogen-orthogonal and orogen-parallel extensional seismicity in the internal part of the Apennines (as discussed earlier in this section) in the overall regional tectonic context of Quaternary orogen orthogonal extension.

**5.4. Active seismotectonics and fault displacement hazard**

The distribution of active seismicity suggests that the MMR is a seismically active region that responds to the extensional tectonic regime currently shaping the internal domain of the Apennines. However, whether this ridge contains active and capable faults remains a matter of debate. This debate has practical implications on the estimate of fault displacement hazard and related seismic risk since several settlements and civil infrastructures are built along and across the fault system.

The IAEA (2010) guidelines propose that, in interplate settings, a fault should be considered as active and capable if it has produced permanent surface deformation in the Upper Pleistocene-Holocene time interval. Machette (2000) recommends that the reliable definition of the state of activity of a fault for seismic hazard assessment should consider a time interval that

includes several earthquake cycles. By applying this concept to the extensional domain of the Apennines, Galadini et al. (2012) propose that a normal fault can be considered as active and capable if it contains evidence of activation since the Middle Pleistocene, unless it is sealed by landforms or deposits older than the Last Glacial Maximum (i.e. ca. 20 ka). The guidelines of the Italian Department for Civil Protection consider a shorter time interval for the definition of active and capable faults, which should be considered such if they display evidence of surface rupture in the last 40 kyr (Technical Commission on Seismic Microzonation, 2015).

The ITHACA catalogue, which applies the definition of active and capable faults proposed by the IAEA (2010) guidelines, considers the Martani Fault System as active and capable by attributing a generic Pleistocene age to its activity (ITHACA Working Group, 2019). However, the supporting evidence to this statement is very dubious since it is based on a large scale neotectonic map of Italy (Ambrosetti et al., 1987b), which impacts its reliability. Based on the interpretation of geomorphic features such as tectonically displaced geomorphic markers, Bonini et al. (2003) suggested that some segments of the MMFS cut through Upper Pleistocene slope debris deposits and displace the Holocene topography, with reactivated N-NNW striking fault segments showing an oblique normal/sinistral kinematics and neoformed NW-SE striking fault segments having normal kinematics, both affecting the Meso-Cenozoic carbonate bedrock. The same study supports these inferences by interpreting ruptures on the *Decumanus* (i.e., a Roman road) in the archeological site of Carsulae as the result of surface faulting during an historical earthquake (around the fifth century AC). If these interpretations are correct, the MMFS should be considered as active and capable. However, subsequent archeological and geophysical studies have confuted this hypothesis by showing that the surface deformation in the Carsulae site is related to the collapse of a doline in the Lower Pleistocene travertine during Roman times, rather than to a seismic event (Aringoli et al., 2009; Bottari & Sepe, 2013; Bottari et al., 2017).

Our field observations document no faulting in deposits younger than the Lower Pleistocene. Where direct relationships between faults and the most recent sedimentary units were observed, faults always appear to be sutured by Upper Pleistocene sediments (Figs. 11, 13 and 15). This implies that the MMFS does not meet the requirements to be considered as active and capable, and such a definition in the public catalogues should, therefore, be reconsidered. This is supported by the fact that the only unquestionable extensional faults cutting the Plio-Quaternary fill of the Medio Tiberino and Terni basins only affect the Upper Pliocene (?) – Lower Pleistocene units (Brozzetti & Lavecchia, 1995; Brozzetti & Stoppa, 1995; Basilici, 1997; this study).

## 6. Concluding remarks

We have shown that the structural grain inherited from the Jurassic pre-orogenic rifting may still represent a major controlling factor on the tectonic evolution of this portion of the Northern Apennines. In fact, although we did not document a reactivation of the western and southern boundaries of the Monti Martani paleo-structural high, the present-day morphostructure of the study area seems to be controlled by the structural framework formed during the Early Jurassic rifting event. This structural inheritance also influenced the structural pattern of the Apennine orogeny in this sector. Most of the post-orogenic extensional

faults that we have documented in this study are broadly orthogonal to the regional extension direction (i.e. NE-SW). By contrast, the inherited ~N-S and ~E-W trends of the pre-orogenic structural grain have a substantial obliquity (~45°) to the regional post-orogenic direction of extension (Fig. 16). This might suggest that for high obliquity between the inherited structures and the regional stress field, the activation of neoformed structures with an optimal orientation with respect to the stress field is mechanically preferred over the reactivation of non-optimally oriented pre-existing faults, as suggested by previous studies, even though the dominant morphostructural trends can still reflect the inherited structural template. This non-optimal orientation between the local structural grain and the far-field stress seems to also favor the formation of fault systems composed of short (few kilometers long) second-order or en echelon faults, rather than the formation of tens of kilometers long faults (Fig. 16). This might have important implications on seismic hazard estimates in this kind of context, since fault length is linearly correlated with the maximum potential magnitude of a seismogenic source.

This work also confirms the importance of field-based structural geology approaches to determine the surface displacement hazard related to complex active fault systems. In fact, a detailed and careful field analysis on single fault segments can confirm or disprove indications that can derive from larger scale studies or geomorphological inferences, which often have a much lower resolution and/or more controversial interpretations. This also makes it possible to gain a better understanding of the evolution of the local stress field through-time, when a long-lasting stable tectonic regime interacts with a non-optimally oriented inherited structural template.

**Author contribution**

*Riccardo Asti*: Conceptualization, Fieldwork, Data collection, Data analysis, Visualization, Writing - original draft, Writing - review & editing. *Selina Bonini*: Fieldwork, Data collection, Writing - review & editing. *Giulio Viola*: Data analysis, Writing - review & editing, Funding acquisition. *Gianluca Vignaroli*: Conceptualization, Fieldwork, Data analysis, Writing - review & editing, Funding acquisition.

**Competing interests**

The authors declare that they have no conflict of interest.

**Acknowledgements**

This work was supported by the PE3 RETURN Project (CUP J33C22002840002; R.A., G. Vign.). We thank Fabrizio Fioroni for providing access to the Grotta Eolia site in Cesi and for his kind and precious assistance during the cave exploration. Valerio Chiaraluce is thanked for stimulating discussions in the archeological site of Carsulae. Simone Fabbi and Costantino Zuccari are thanked for helpful discussions about the pre-orogenic inheritance in the Apennine system. We thank Gianluca

Benedetti, Massimo Comedini, Stefano Rodani and Giulia Tartaglia for stimulating discussions in the field and for encouraging us throughout the realization of this study. We thank topic editor Stefano Tavani for the editorial handling of this work. This

manuscript benefited from thoughtful comments and suggestions by Marco Mercuri and an anonymous reviewer.

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
