# Peer review of "Reconciling post-orogenic faulting, paleostress evolution and structural inheritance in the seismogenic Northern Apennines (Italy): Insights from the Monti Martani Fault System"

_EGUsphere, 2024_

## Referee Comment (RC1)

**Revision of "Reconciling post-orogenic faulting, paleostress evolution and structural inheritance in the seismogenic Northern Apennines (Italy): Insights from the Monti Martani Fault System" by Riccardo Asti and coauthors**

**General Comments**

In this study, Asti and co-authors present a structural analysis of a fault system in central Italy. Specifically, the study focuses on the Monti Martani Fault System (MMFS), which forms the western and southern boundaries of the Monti Martani Ridge (MMR). This ridge is elongated in a North–South direction and is characterized by Mesozoic structural highs and lows that were later involved in the Apennine orogeny. The MMFS is analyzed through very precise mesoscale observations, and the collected structural data are used to reconstruct the paleo–stress field and to improve the fault system mapping.

Asti and co-authors demonstrate that the recent extensional fault system is segmented and consists of fault segments oriented NW–SE and, to a lesser extent, NE–SW. These observations contradict previous literature, which described the MMR as being bordered by a single, L–shaped fault. Asti and coauthors therefore show that the morphostructure of the MMR, elongated in a N–S direction, is not exclusively controlled by the recent extensional fault system. Instead, the elongation of the MMR more closely aligns with the orientation of the older Jurassic faults that bordered the structural highs. The authors propose that the architecture of the MMFS, particularly the distribution of its segments, is controlled by the orientation of these older Jurassic faults, even though they were not reactivated.

The inversion of kinematic data suggests three extensional regimes with different orientations of the principal minimum stress: NE–SW, NNE–SSW, and NW–SE. The first two are attributable to the recent extensional phase, while the latter is subsequent to the first two and is more difficult to interpret. In my opinion, the structures associated with a NW–SE extensional field are too localized (2 sites on a structure that extends for tens of kilometers) to be attributed to a regional stress field. These same structures could be interpreted as local complications in the stress field. Therefore, I believe this more conservative interpretation should be preferred.

Overall, I find this study highly interesting for the readers of Solid Earth as an example of the application of structural observations to better constrain the architecture of an extensional fault system. In particular, I believe it could serve as a valuable case study for a better understanding of the role of inherited structures in the development of recent extensional systems. However, before the paper can be considered for publication, I suggest that the paleogeography and,

consequently, the orientation of the Jurassic extensional faults in the Monti Martani be better described, particularly concerning Figure 3. In this Figure, the Meso-Cenozoic succession is represented in a uniform color without differentiating formations and key elements, which would be necessary to allow readers to infer the existence and orientation of the Jurassic structural high(s).

Please find below some specific comments.

Best regards,
Marco Mercuri

**Specific comments**

**Abstract**. The methods section of the abstract is somewhat vague and lacks detail: please briefly specify the scale at which the observations were made, and the type of analysis conducted. The aim of the study is clearly stated and focuses on using the MMFS as a case study to understand the role of pre-orogenic inherited structures in post-orogenic tectonic evolution.

- **Line 20 and following**: The relationship between the paleostress field analysis and the morphological control of the N-S and E-W pre-orogenic faults is not immediately clear.

**Introduction**. The relevance (Lines 30-34) and the aim (Lines 53-57) are clearly stated. Please consider the following suggestions:

- **Line 30:** In complex fault systems, the orientation and spatial distribution of the individual fault segments
- **Lines 30 - 34:** The conceptual link between these two sentences is unclear. I suggest stating that structural inheritance influences the architecture of the fault system (i.e., the spatial distribution, orientation and linkage between different slip surfaces at a 100m-1km scale). Consequently, structural inheritance also plays a role in the seismological behavior of the fault system.
- **Lines 43-45:** For better clarity, I suggest explicitly stating that the ~N-S structural features separate different paleogeographic domains before the orogeny.

**Geological setting.** The presence of a structural high during the Mesozoic is very important for the aims of this work. This is well described in the Geological Setting (Lines 9-123) but it is not

adequately reflected in the figures. The main issue is that the Meso-Cenozoic succession is not differentiated in Figures 3 and 4. In my opinion, a more detailed mapping would improve the manuscript. A good compromise would be to differente the pre-rift (i.e., Calcare Massiccio), the condensed pelagic succession, the complete pelagic succession, and the Maiolica-Schlier interval.

- **Line 78**: I suggest using the term "intrabasinal structural highs" because the Apenninic platform is also a structural high and does not have a condensed sequence on its top.
- **Line 80 and following:** The strike of the Jurassic faults appears to be quite important in your study, but it is not mentioned here. I suggest adding a sentence about it.
- **Line 108:** From the earlier description, it seems that the MMFS separates the Medio Tiberino and Terni basins (Line 49). I suggest adding these names in Figure 1b for better clarity.
- **Figure 3:** Please consider indicating the difference between active and inactive alluvial fans.

**Methods:** I believe the methods are adequate for characterizing the architecture of the fault system and for the analysis of the paleostress. However, I am not sure if the applied methods are sufficient to investigate the structural inheritance of the pre-orogenic structures on the post-orogenic extensional fault system (see also previous point). Please consider also the following suggestions:

- **Figure 5:** The text within this figure is difficult to read (one has to zoom in at 200% to read it). Please consider increasing the font size slightly. I also suggest explaining the "gaussian parameters" in the caption and adding bedding information to each stereoplot, if possible.
- **Figure 6:** To improve readability, I suggest adding a miniature of Figure 3 somewhere in the figure, showing the location of the structural stations along the MMFS.

**Results**: In many figures showing field observations (e.g., Figs. 8-15), it is difficult to see the structures described in the text (e.g., the NE-SW striking veins in Figure 8) because they are obscured by the line drawings. I suggest finding a way to make these structures more visible in the figures. Additionally, the need to constantly switch between the field observation figures (8-15), the figure showing the location of the structural station (3), the stereoplots (6), and the kinematic inversion (7) makes the text somewhat difficult to follow. I suggest reorganizing the figures to make the text easier to follow.

- **Figure 5c:** To better visualize the three main azimuths of the slickenlines, I suggest, if possible, showing them in a rosediagram (similar to Figure 5a).
- Regarding the NE-SW oriented veins described in the Viepri site, which cut the cataclasite and the hanging wall of the fault: are they also present in the footwall? I agree that they

may form in a stress field with NW–SE minimum principal stress, but at the same time, I think this could be a local complication of the stress field. For example, a variation in throw along the strike of the main fault could be accommodated by extensional structures perpendicular to it (e.g., release faults; Destro, 1995).

- **Line 321**: I am not sure it can be called a "neptunian dyke" if the sedimentary infill occurs outside the submarine environment. I suggest double-checking the use of this term.
- **Figure 14:** Dip-slip slickenlines are visible, while oblique-slip and strike-slip slickenlines are very difficult to discern. Perhaps better image quality could help.
- There are a lot of figures. Consider moving some to supplementary material, if possible.
- In many figures showing field observations, the line drawings obscure the visibility of the structures themselves.

**Discussion.** In my opinion, the section on the structural inheritance of Jurassic faults in the morphostructure of the MMR (section 5.2) is very interesting. However, I believe it needs more "visual" support regarding the paleomorphology of the Jurassic structural high. Specifically, I think it is necessary to at least hypothesize the location and the orientation of the paleoescarpments. One idea could be to provide more detail in the map in Figure 3 by differentiating the Meso-Cenozoic succession into Calcare Massiccio condensed succession, complete succession, and post-paleotopography succession (i.e., from Maiolica) (see also the comment in the Geological Setting). The visualization of a transition from a structural high to a structural low proceeding in the footwall towards the MMFS could be helpful. The section concerning the existence of a NW–SE extensional stress field (section 5.3) is supported by limited data: the subhorizontal slickenlines younger than dip-slip ones on NW–SE oriented fault planes are convincing, but besides being well documented only in the Cesi quarry, they could also be attributed to local complications of the stress field. Local complications of the stress field might occur, for example, due to interaction between the main fault segments. The veins that cut the cataclasite and the hanging wall of the fault near the Viepri site could also be interpreted as a very local complication of the stress field (see comments in the Result section). I believe the interpretation of the structural data should also consider the possibility that a regional NW–SE extensional stress did not exist.

- **Lines 410-413**: I find the alignment of the epicenters in the NW–SE or WNW–ESE direction difficult to identify. My suggestion is to make this more visible in some way in Figure 3, or consider removing the sentence.
- **Lines 485-490 & 498-528**: I believe that the NW–SE extensional stress field, being "documented" in only two locations along a structure that extends for tens of kilometers (as shown in Figure 17), cannot be attributed to tectonics on the scale of the structure

itself or larger. I think the more conservative explanation is the existence of a local NW–SE extension direction in areas with local structural complications (e.g., relay zones?).

**Conclusions**

- **L 574-576**: What do you mean by "kilometer-long"? Did you perhaps mean "tens of kilometers"?

**Technical corrections**

- L. 38: siyn-orogenic
- L. 49 forming a ~N-S structural ridge
- L. 49: I suggest highlighting the Medio Tiberino and Terni basins in Figure 1b.
- L. 51:  from
- L. 150  and
- Line 189. The web address provided does not work. Please update the link to the correct website.
- Figure 7. What do the different colors of the arrows represent? For example, what is the significance of the purple versus black arrows in panel c? Please explain the symbols in the caption.
- Figure 8. Please, find a way to make the veins visible also without the line-drawing obscuring them.
- Figure 9. I assume S0 represents bedding, but it is not mentioned in the caption
- L.278. Stereonet n. 8 in Fig. **6**
- L.290. Did you mean "fault core"? In my opinion cataclasite implies rotation and disaggragation of grains which makes it impossible to see the primary features (whose recognition allows to identify a damage zone).

---

## Author Comment (AC1)

*General Reply*

*Dear Dr. Marco Mercuri,*

*We wish to thank you for your thorough review and insightful comments on our manuscript entitled "Reconciling post-orogenic faulting, paleostress evolution and structural inheritance in the seismogenic Northern Apennines (Italy): Insights from the Monti Martani Fault System". We sincerely appreciate the time and the effort that you spent to give a constructive contribution to our work. Your comments and suggestions have been thoroughly considered and integrated in the revised version of the manuscript. This surely helped us to improve the quality of the manuscript.*

*Here below, we respond to each of your comments. We report your original comments in* **black***, followed by the relative responses in* **red***. We are confident that the suggested adjustments will strengthen the overall quality of the manuscript and we hope that our replies to your comments and the way we integrated them in the revised version of the manuscript will meet Your satisfaction.*

*Yours sincerely,*

*Riccardo Asti*

*(on behalf of the co-authors)*

**General Comments**

In this study, Asti and co-authors present a structural analysis of a fault system in central Italy. Specifically, the study focuses on the Monti Martani Fault System (MMFS), which forms the western and southern boundaries of the Monti Martani Ridge (MMR). This ridge is elongated in a North-South direction and is characterized by Mesozoic structural highs and lows that were later involved in the Apennine orogeny. The MMFS is analyzed through very precise mesoscale observations, and the collected structural data are used to reconstruct the paleo-stress field and to improve the fault system mapping.

Asti and co-authors demonstrate that the recent extensional fault system is segmented and consists of fault segments oriented NW-SE and, to a lesser extent, NE-SW. These observations contradict previous literature, which described the MMR as being bordered by a single, L-shaped fault. Asti and coauthors therefore show that the morphostructure of the MMR, elongated in a N-S direction, is not exclusively controlled by the recent extensional fault system. Instead, the elongation of the MMR more closely aligns with the orientation of the older Jurassic faults that bordered the structural highs. The authors propose that the architecture of the MMFS, particularly the distribution of its segments, is controlled by the orientation of these older Jurassic faults, even though they were not reactivated.

The inversion of kinematic data suggests three extensional regimes with different orientations of the principal minimum stress: NE-SW, NNE-SSW, and NW-SE. The first two are attributable to the recent extensional phase, while the latter is subsequent to the first two and is more difficult to interpret. In my opinion, the structures associated with a NW-SE extensional field are too localized (2 sites on a structure that extends for tens of kilometers) to be attributed to a regional stress field. These same structures could be interpreted as local complications in the stress field. Therefore, I believe this more conservative interpretation should be preferred.

Overall, I find this study highly interesting for the readers of Solid Earth as an example of the application of structural observations to better constrain the architecture of an extensional fault system. In particular, I believe it could serve as a valuable case study for a better understanding of the role of inherited structures in the development of recent extensional systems. However, before the paper can be considered for publication, I suggest that the paleogeography and, consequently, the orientation of the Jurassic extensional faults in the Monti Martani be better described, particularly concerning Figure 3. In this Figure, the Meso-Cenozoic succession is represented in a uniform color without differentiating formations and key elements, which would be necessary to allow readers to infer the existence and orientation of the Jurassic structural high(s).

**REPLY:** Replies to the points raised here above by the Reviewer are extensively discussed here below in the responses to the specific comments.

**Specific comments**

**Abstract.** The methods section of the abstract is somewhat vague and lacks detail: please briefly specify the scale at which the observations were made, and the type of analysis conducted.

**REPLY:** This information has been added to the part of the text dealing with the methods. In particular, we now clarify that observations and structural measurements were made at the outcrop scale and that the collected data (geometry and kinematic fault-slip data) were used to perform paleostress analysis.

The aim of the study is clearly stated and focuses on using the MMFS as a case study to understand the role of pre-orogenic inherited structures in post-orogenic tectonic evolution.

• Line 20 and following: The relationship between the paleostress field analysis and the morphological control of the N-S and E-W pre-orogenic faults is not immediately clear.

**REPLY:** To better clarify this point, we modified this part of the manuscript to specify that the orientation of the morphostructure is independent from the post-orogenic paleostress field and is rather controlled by inherited (pre-orogenic) structural grain.

**Introduction.** The relevance (Lines 30-34) and the aim (Lines 53-57) are clearly stated. Please consider the following suggestions:

• Line 30: In complex fault systems, the orientation and spatial distribution of the individual fault segments

**REPLY:** This change has been added to the text.

• Lines 30 – 34: The conceptual link between these two sentences is unclear. I suggest stating that structural inheritance influences the architecture of the fault system (i.e., the spatial distribution, orientation and linkage between different slip surfaces at a 100m-1km scale). Consequently, structural inheritance also plays a role in the seismological behavior of the fault system.

**REPLY:** The first paragraph of the Introduction has been modified according to this comment.

• Lines 43-45: For better clarity, I suggest explicitly stating that the ~N-S structural features separate different paleogeographic domains before the orogeny.

**REPLY:** This modification has been added to the text.

**Geological setting.** The presence of a structural high during the Mesozoic is very important for the aims of this work. This is well described in the Geological Setting (Lines 9-123) but it is not adequately reflected in the figures. The main issue is that the Meso-Cenozoic succession is not differentiated in Figures 3 and 4. In my opinion, a more detailed mapping would improve the manuscript. A good compromise would be to differente the pre-rift (i.e., Calcare Massiccio), the condensed pelagic succession, the complete pelagic succession, and the Maiolica-Schlier interval.

**REPLY:** We have drawn a new version of the Figure 3 where, following the requests and suggestions made by both reviewers, we try to highlight the distribution of the Jurassic condensed succession to underline the presence of Jurassic paleo-escarpments. This new version of the figure will be included in the revised version of the manuscript.

• Line 78: I suggest using the term "intrabasinal structural highs" because the Apenninic platform is also a structural high and does not have a condensed sequence on its top.

**REPLY:** Done.

• Line 80 and following: The strike of the Jurassic faults appears to be quite important in your study, but it is not mentioned here. I suggest adding a sentence about it.

**REPLY:** Done.

• Line 108: From the earlier description, it seems that the MMFS separates the Medio Tiberino and Terni basins (Line 49). I suggest adding these names in Figure 1b for better clarity.

**REPLY:** These names have been added in the revised version of this figure.

**Methods:** I believe the methods are adequate for characterizing the architecture of the fault system and for the analysis of the paleostress. However, I am not sure if the applied methods are sufficient to investigate the structural inheritance of the pre-orogenic structures on the postorogenic extensional fault system (see also previous point). Please consider also the following suggestions:

• Figure 5: The text within this figure is difficult to read (one has to zoom in at 200% to read it). Please consider increasing the font size slightly. I also suggest explaining the "gaussian parameters" in the caption and adding bedding information to each stereoplot, if possible.

**REPLY:** The size of the text has been increased in the revised version of the figure and the gaussian parameters are now explained in the figure caption in the revised version of the manuscript.

• Figure 6: To improve readability, I suggest adding a miniature of Figure 3 somewhere in the figure, showing the location of the structural stations along the MMFS.

**REPLY:** We thank the reviewer for this suggestion. However, we prefer not to add a map to this figure, because it would take too much space in the page and it will necessarily decrease the size of the stereonets, making them hardly readable.

**Results:** In many figures showing field observations (e.g., Figs. 8-15), it is difficult to see the structures described in the text (e.g., the NE-SW striking veins in Figure 8) because they are obscured by the line drawings. I suggest finding a way to make these structures more visible in the figures. Additionally, the need to constantly switch between the field observation figures (8-15), the figure showing the location of the structural station (3), the stereoplots (6), and the kinematic inversion (7) makes the text somewhat difficult to follow. I suggest reorganizing the figures to make the text easier to follow.

• Figure 5c: To better visualize the three main azimuths of the slickenlines, I suggest, if possible, showing them in a rosediagram (similar to Figure 5a).

**REPLY:** A rose diagram displaying these data has been added to Figure 5c in the revised version of the figure.

• Regarding the NE-SW oriented veins described in the Viepri site, which cut the cataclasite and the hanging wall of the fault: are they also present in the footwall? I agree that they may form in a stress field with NW-SE minimum principal stress, but at the same time, I think this could be a local complication of the stress field. For example, a variation in throw along the strike of the main fault could be accommodated by extensional structures perpendicular to it (e.g., release faults; Destro, 1995).

**REPLY:** Based on what we could observe in the field, these veins are not found in the footwall of the faults in the Viepri site. We cannot totally exclude their presence, due to limited exposures of the footwall rocks in the area. They have been observed only cutting through the cataclasite of the fault core and through the Cenozoic marls in the hanging wall. Although these veins have the same orientation and might look similar at the outcrop scale, preliminary observations show that there are microtextural differences between them.

Concerning the tectonic interpretation of these veins, please see the reply below to the comment to the Discussion section.

• Line 321: I am not sure it can be called a "neptunian dyke" if the sedimentary infill occurs outside the submarine environment. I suggest double-checking the use of this term.

**REPLY:** We agree with the Reviewer that our use of the term "neptunian dike" was inappropriate. We replaced it with the more appropriate "fissure infill".

• Figure 14: Dip-slip slickenlines are visible, while oblique-slip and strike-slip slickenlines are very difficult to discern. Perhaps better image quality could help.

**REPLY:** In the revised version of the manuscript, we increased the resolution of the photos showing the slickenlines. They should be now much more visible.

• There are a lot of figures. Consider moving some to supplementary material, if possible.

**REPLY:** We thank the Reviewer for this suggestion. However, since field observations are a relevant part of our dataset and we do not find that there are redundant figures, we prefer to keep them in the main manuscript, rather than moving them to the supplementary material.

• In many figures showing field observations, the line drawings obscure the visibility of the structures themselves.

**REPLY:** We agree with this comment. In order to make the structures more visible in the field pictures, we decreased the size of the line drawings on the figures.

**Discussion.** In my opinion, the section on the structural inheritance of Jurassic faults in the morphostructure of the MMR (section 5.2) is very interesting. However, I believe it needs more "visual" support regarding the paleomorphology of the Jurassic structural high. Specifically, I think it is necessary to at least hypothesize the location and the orientation of the paleoescarpments. One idea could be to provide more detail in the map in Figure 3 by differentiating the MesoCenozoic succession into Calcare Massiccio condensed succession, complete succession, and post-paleotopography succession (i.e., from Maiolica) (see also the comment in the Geological Setting). The visualization of a transition from a structural high to a structural low proceeding in the footwall towards the MMFS could be helpful.

**REPLY:** Figures 3 and 4 have been modified according to this comment (see reply to the comment to the Geological Setting section).

The section concerning the existence of a NWSE extensional stress field (section 5.3) is supported by limited data: the subhorizontal slickenlines younger than dip-slip ones on NW-SE oriented fault planes are convincing, but besides being well documented only in the Cesi quarry, they could also be attributed to local complications of the stress field. Local complications of the stress field might occur, for example, due to interaction between the main fault segments. The veins that cut the cataclasite and the hanging wall of the fault near the Viepri site could also be interpreted as a very local complication of the stress field (see comments in the Result section). I believe the interpretation of the structural data should also consider the possibility that a regional NW-SE extensional stress did not exist.

**REPLY:** We agree with the Reviewer that a regional NW-SE extensional stress cannot be the only possible explanation for the structures observed in the Viepri site (site 1), in the Cesi quarry (site 10b) and east of Cesi (site 9). In fact, as suggested by the Reviewer, these could also be related to local perturbations of the stress field and represent minor structures that accommodate variations in the throw rate along major structures that are optimally oriented with respect to the regional stress field. However, in the Discussion section in the originally submitted version of the manuscript, we did not make any choice on the preferred possible interpretation of these structures. We find important to discuss the possibility that these structures might have some relevance with respect to the regional evolution of the post-orogenic tectonic regime, since analogue observations have been made in analogue contexts along the Apennines belt (as discussed in the manuscript). However, we agree that we did not clarify enough that an explanation of regional relevance was not the only possible explanation, so we modified this part of the manuscript adding references to the possibility that these structures might also result from local perturbations of the stress field during fault evolution.

• Lines 410-413: I find the alignment of the epicenters in the NW-SE or WNW-ESE direction difficult to identify. My suggestion is to make this more visible in some way in Figure 3, or consider removing the sentence.

**REPLY:** This sentence has been modified according to the Reviewer's comment.

• Lines 485-490 & 498-528: I believe that the NW-SE extensional stress field, being "documented" in only two locations along a structure that extends for tens of kilometers (as shown in Figure 17),

cannot be attributed to tectonics on the scale of the structure itself or larger. I think the more conservative explanation is the existence of a local NWSE extension direction in areas with local structural complications (e.g., relay zones?).

**REPLY:** In agreement with the Reviewer's comment, we modified this part of the manuscript and we concluded this section stating that the orogen-parallel extensional stress field might result from local perturbations of the regional stress field (see also reply to previous comment), making reference to classical examples of relay zones in extensional systems.

**Conclusions**

• L 574-576: What do you mean by "kilometer-long"? Did you perhaps mean "tens of kilometers"?

**REPLY:** Yes, exactly. We corrected this sentence in the revised version of the manuscript.

**Technical corrections**

• L. 38: siyn-orogenic

**REPLY:** This correction was made in the revised manuscript.

• L. 49 forming a ~N-S structural ridge

**REPLY:** This correction was made in the revised manuscript.

• L. 49: I suggest highlighting the Medio Tiberino and Terni basins in Figure 1b.

**REPLY:** The figure has been modified accordingly.

• L. 51: form from

**REPLY:** This correction was made in the revised manuscript.

• L. 150 ang and

**REPLY:** This correction was made in the revised manuscript.

• Line 189. The web address provided does not work. Please update the link to the correct website.

**REPLY:** Instructions on how to obtain the software have been updated in the revised manuscript.

• Figure 7. What do the different colors of the arrows represent? For example, what is the significance of the purple versus black arrows in panel c? Please explain the symbols in the caption.

**REPLY:** In Figure 7, black arrows represent measured slip vector on a fault plane, while purple arrows represent the theoretical optimal slip vector on a fault plane with respect to the computed paleostress tensor. This information has been added in the caption of Figure 7.

• Figure 8. Please, find a way to make the veins visible also without the line-drawing obscuring them.

**REPLY:** This figure has been modified according to this comment in order to increase the visibility of the veins.

• Figure 9. I assume S0 represents bedding, but it is not mentioned in the caption

**REPLY:** S0 has been substituted with "bedding" in the revised version of this figure.

• L.278. Stereonet n. 8 in Fig. 6

**REPLY:** This correction was made in the revised manuscript.

• L.290. Did you mean "fault core"? In my opinion cataclasite implies rotation and disaggragation of grains which makes it impossible to see the primary features (whose recognition allows to identify a damage zone).

**REPLY:** To avoid misunderstanding, we deleted "of the damage zone".

---

## Author Comment (AC2)

*General Reply*

*Dear Reviewer,*

*We wish to thank you for your thorough review and insightful comments on our manuscript entitled "Reconciling post-orogenic faulting, paleostress evolution and structural inheritance in the seismogenic Northern Apennines (Italy): Insights from the Monti Martani Fault System". We sincerely appreciate the time and the effort that you spent to give a constructive contribution to our work. Your comments and suggestions have been thoroughly considered and subsequently integrated in the revised version of the manuscript. This surely helped us to improve its quality.*

*Here below, we respond to each of your comments. We report your original comments in* **black***, followed by the relative responses in* **red***. We also attach an annotated pdf where we respond to the specific comments and typographic/editorial suggestions. We are confident that the suggested adjustments will strengthen the overall quality of the manuscript. As you will notice in this rebuttal letter, we do not agree with all of your comments and we tried to clarify our position where our views differ from yours.*

*Yours sincerely,*

*Riccardo Asti*

*(on behalf of the co-authors)*

**General Comments**

In this study, the Authors present field data from structural stations to analyze the influence of pre-orogenic tectonic lineaments on the development of post-orogenic normal faults. The research was conducted in a distinctive area of the Central Apennines, specifically the Martani Mountains. This region, like the entire Umbria-Marche-Sabina Apennine, has undergone polyphase tectonic events recorded in the Meso-Cenozoic shallow-water to pelagic carbonates, Miocene syn-orogenic terrigenous units, and post-orogenic Plio-Quaternary marine-to-continental deposits. The study emphasizes the role of Tethyan rift inheritance, proposing that the present-day ridge geometry is controlled by these rift-related faults. According to the authors, these faults notably influenced the morphostructural configuration of the western and southern sectors of the Martani Mountains, resulting in an L-shaped ridge morphology. However, the authors did not perform detailed field mapping aimed at reconstructing the Jurassic paleotectonic setting or pre-orogenic tectonic architecture of the study area, instead inferring this from the existing literature.

The realization of structural stations at key outcrops and the subsequent analysis of the collected data led the authors to identify a set of several NW-SE trending extensional faults (defined "short"), arranged in an en-echelon pattern. These faults contrast with the N-S and WNW-ESE trending faults that bound the Martani Mountains to the west and south, which have been widely discussed in the literature. The field structural data were further analyzed to reconstruct the paleostress regime, revealing three distinct extension directions—NE-SW, NNE-SSW, and NW-SE—attributed to the Plio-Quaternary post-orogenic extension.

The identification of faults cutting through Pliocene and, in part, Pleistocene deposits, yet sealed by Upper Pleistocene deposits, allowed the authors to infer their potential seismotectonic significance.

In my opinion, the manuscript is written in clear and proficient English. The paper holds significant scientific potential and could be of interest to the Solid Earth audience. However, it has several weaknesses that should be addressed before it can be considered for publication. In particular, I recommend that the Authors:

- Provide clearer and more substantial field evidence to support their inferences regarding the existence of Plio-Quaternary faults intersecting and downthrowing high-displacement, pre-existing normal faults.

- Better define the pre-orogenic paleotectonic and stratigraphic framework of the study area, as well as the Plio-Pleistocene one.

- Strengthen the field constraints that underpin speculations about seismic hazard and paleostress analysis. This can only be achieved after establishing a relative chronology of the analyzed faults. As a result, intraformational faults should not be included in the analysis.

Specific comments are extensively discussed in the attached PDF, where editorial and typographic corrections are also noted. Additionally, several important citations of studies focused on the Martani Mountains are missing.

**REPLY:** Replies to the points raised here above by the Reviewer are extensively discussed here below in the responses to the specific comments and in the attached annotated pdf version of the originally submitted manuscript.

**Specific comments**

***Early Jurassic paleotectonic-stratigraphic architecture of the Martani Mts.***

The authors base their structural inferences on the influence of Early Jurassic rift faults in the development of pre-orogenic(?)/syn-orogenic(?)/post-orogenic(?) normal faults bounding the Martani Mountains Ridge (MMR), which result in its L-shaped morphology. However, the distinction between these stages is unclear in the text (see comments below). According to the Authors, the kilometers-long N-S and WNW-ESE trending fault(s) bounding the MMR to the west and south (i.e., Martana Fault *Auctt*) are Jurassic inheritances, as already stated by Bruni et al. (1995) and Cipriani et al. (2020). Nonetheless, the Authors do not provide their own field data to support this inference, which becomes evident from figures 3 and 4.

In fact, the geological map reported in Figure 3 is inadequate, as it is derived from a study that does not take the pre-Quaternary stratigraphy into account at all. By contrast, a study where structural inferences are drawn from inherited Jurassic paleotectonic and stratigraphic architecture requires a geological map that clearly delineates the pre-rift bedrock, the facies and thickness variations of syn and post-rift deposits, which are characteristic of the study region. Additionally, indirect evidence of rift faults - represented in the area by unconformity surfaces rather than shear zones (see Santantonio et al., 2017 for further details) - is missing, representing a major gap in the data. Comparable features are in Figure 4, where an oversimplified and obsolete litho-chronostratigraphic setting has been reported.

I suggest providing field evidence the pre-orogenic paleotectonic and stratigraphic framework of the study area, that could be identified not in limited structural stations, but after a widespread geological mapping involving the whole area.

**REPLY:** We thank the Reviewer for this comment, which provides us with the opportunity to further clarify the aims of our work and address some misunderstandings. Our intention was not to prove and document the existence of pre-orogenic structures (which are already documented in the literature) and their control on the tectonic evolution of the study area. Our goal was instead to study the structural relationships between the Monti Martani Ridge (MMR) and the adjacent Terni and Medio Tiberino Basins. To this end, we collected fault slip data from all along the trace of the "so-called" Martana Fault, i.e., from along the western and southern boundaries of the MMR. There are fundamental misfits between the structural data along their boundary and their "classical" interpretation in the literature (this was clearly stated in the original manuscript – see for example lines 12-15, 136-144, 401-405). In fact, if one looks at the structural data collected all along the trace of the "so-called" Martana Fault from the literature (i.e., Barchi et al., 1991; Brozzetti & Lavecchia, 1995; Bonini et al., 2003) and that are corroborated by our new original data, one will notice that the vast majority of these data confirm the presence of NW-SE striking faults along the MMR/basins boundary, rather than N-S or WNW-ESE trending faults, which should instead exist if the Martana Fault morphostructural boundary truly existed with the geometry that is commonly reported in the literature. This more complex structural architecture was already recognized about 50 years ago by some authors, who showed that the western boundary of the MMR is bounded by a several tens of kilometers long fault system composed of 2-3 kilometers long fault segments with a left-stepping en-echelon geometry, with a complex distribution of minor vicarious dislocations (Giglia et al., 1977, according to Barchi et al., 1991).

However, for some mysterious reasons, the idea of an L-shaped N-S to WNW-ESE trending Martana Fault became dominant in the literature in the following decades. So, in our view, as is also explicitly stated in the manuscript, "the kilometer-long N-S and WNW-ESE trending fault(s) bounding the MMR to the west and south (i.e., Martana Fault *Auctt*)" probably does not exist in the form in which it is commonly presented in the literature, or at least there is not a single evidence for its existence that can be observed at the surface. In fact, fault data collected along the western and southern margins of the MMR presented in this study and in the literature (Barchi et al., 1991; Brozzetti & Lavecchia, 1995) show that the vast majority of the extensional faults strike NW-SE.

So, in summary, our work attempts at shedding some light on this controversial issue by strengthening the structural dataset available to the community and making inferences as to why this odd situation may have arisen. It was not our goal to "map the whole area", as the reviewer suggests, nor to specifically focus on stratigraphic and structural details concerning the Mesozoic rifting history.

What we propose in the manuscript is rather that the morphostructure of the MMR might somehow be controlled by the pre-orogenic inherited structural grain, and do not claim that the MMR is bounded by N-S and WNW-ESE striking fault(s). Despite the evident mismatch between the existing structural data and the existence of a supposed N-S to WNW-ESE trending Martana Fault (generally accepted by other authors), this inconsistency has never been addressed up to now.

Thus, our goal was to explicitly address this problem and this problem only by presenting systematic meso-scale structural observations along the morphostructural boundary between the MMR and the Plio-Quaternary basins. Only after discussing the data, we propose the role of the pre-orogenic structural inheritance as a potential controlling factor for the present-day morphology of the MMR and for the structural evolution of the ridge/basins boundary. What happens to faults that we (and other studies) documented along this boundary once these intersect pre-existing faults in more internal part of the MMR remains an open question that is beyond the scope of this paper and that we, and/or other researchers, may want to develop in future studies.

We want this point to be very clear as it obviously impacts significantly on many of the comments that we have received from both reviewers, which, in light of this further explanation, are not truly central to our study, its novelty and the specific impact it may have.

Throughout the review, it is rather evident that there is a fundamental misunderstanding by the Reviewer (see also the replies to many of the following comments) on the difference between what is an "inherited structure" or an "inherited fault" and what is an "inherited structural grain" (term that we rather use throughout the manuscript). The "inherited structural grain" can be defined as the summation of stratigraphic, geometric, kinematic and tectonic features that, as a whole, express the bulk anisotropy of the crustal block undergoing subsequent deformation. Thus, one should not necessarily look for a Jurassic fault plane that may have been reactivated. One anisotropy (or a set of them) of any kind and origin may be sufficient to have an impact on subsequent deformation events. In order to avoid that this misunderstanding arises in other readers, we will clarify this definition in the revised version of the manuscript.

Consequently, I invite to re-edit the geological map using the geological cartography of Regione Umbria at 1:10,000 scale, which should be updated with respect to the official geological map of Italy at 1:100,000 scale. Furthermore, I suggest to differentiate at least:

-the Calcare Massiccio, the latter being the pre-rifting substrate and providing information about the paleotectonic and stratigraphic setting;

- the Jurassic basinal succession, providing at least a unique color for the Corniola to Calcari Diasprigni formations;

- the Jurassic PCP-top condensed succession of the Bugarone group;

- the Maiolica to Bisciaro carbonate succession;

- the Schlier and Marnoso-Arenacea formations as siliciclastic succession.

**REPLY:** All what we rebutted immediately above notwithstanding, we are prepared to improve the manuscript to the best of our possibilities. This is why we are drawing a new version of the Figure 3 where, following the requests and suggestions made by both reviewers, we try to highlight the distribution of the Jurassic condensed succession to underline the presence of Jurassic paleo-escarpments. This new version of the figure will be included in the revised version of the manuscript.

Analogously, I strongly recommend replacing the chrono-lithostratigraphic scheme in Figure 4 with the updated version presented in Curzi et al. (2024), or at least drawing inspiration from it. The co-authors of the current work are also co-authors of the previously cited paper. Additionally, the colors used for the stratigraphic intervals should align with those specified earlier for the geological map.

**REPLY:** We thank the Reviewer for this suggestion. This figure will be updated in the revised version of the manuscript according to the Reviewer's comment.

*Role of Jurassic rifting faults and present-day geometry*

Assuming that the Martana Fault System is a Jurassic inheritance, and granting that I can also agree with what the Authors report, however, it is not clear in the text:

**REPLY:** Here, again, there is a misunderstanding of our statements. In the manuscript, we never say that the Martana Fault System directly stems from the reactivation of a specific Jurassic inherited fault. According to the literature, most of the Jurassic faults strike N-S and E-W. Rather, we show that the Monti Martani Fault System is composed of short and disconnected NW-SE striking normal fault segments broadly distributed along the morphostructural boundary between the MMR and the adjacent basins (which trends N-S and WNW-ESE). Thus, we do not propose that this structural architecture results from the direct reactivation of Jurassic faults, but rather that post-orogenic faulting was influenced by the Jurassic inherited and complex structural template.

- if the fault system acted only in the pre-orogenic stage (i.e., since the Jurassic up to the Early Miocene);

  **REPLY:** If the Reviewer here refers to the Jurassic faults, this is not something that we have investigated, and it goes beyond the scope of our work. To our interpretation, what counts is the geometry of the bulk inherited structural grain and its orientation compared to the post-orogenic extensional stress field. We do not focus the attention on their potential reactivation before the Plio-Quaternary extension. It is very likely that the Jurassic structural grain reactivated and/or influenced the structural expression of the tectonic events that preceded the post-orogenic phase (as suggested by previous studies - see for example: Bruni et al., 1995; Bonini, 1998), but this is not the core of our interpretation. However, even though we did not document reactivation of specific pre-orogenic structures in our study, considering the orientation of the paleostress tensor related to the post-orogenic extension (i.e., $\sigma_3$ mainly NE-SW oriented), inherited N-S striking faults should have reactivated as sinistral oblique transtensional faults, while ~E-W striking faults should have reactivated as dextral oblique transtensional faults during this event, as also proposed by previous studies (e.g., Brozzetti & Lavecchia, 1995; Bonini et al., 2003).

- if if the fault system was reactivated in the Plio-Quaternary times, as stated for instance in the lines 430-435.;

  **REPLY:** In this part of the manuscript (nor elsewhere) we don't state that the Plio-Quaternary extension reactivates Jurassic faults. Again, we think there is a fundamental misunderstanding between what we define as the Monti Martani Fault System and what are the Jurassic faults.

- if the fault system was dissected by Plio-Quaternary NW-trending faults. In fact, the same Authors provide a structural dataset that is in contrast with what said above. According with them, the faults analysed in this manuscript are not coherent with the Plio-Quaternary regional stress direction and interacted with those inherited from the pre-orogenic phase. In the "Abstract" section the Authors write (Lines 12-15): "*Based on new field structural data from extensional faults that controlled the Plio-Quaternary evolution of the system, we propose that the MMFS does not consist of a kilometer-long L-shaped single normal fault, as previously proposed in the literature, but is instead a set of several NW-SE trending shorter extensional faults arranged in an en-echelon style.*" Analogously, in the lines 401-405: "*…the morphostructural trend of the western and southern margins of the MMR does not correspond to continuous and several kilometer-long fault traces aligned along the ~N-S or WNW-ESE directions. Rather, the MMFS appears to be formed of several disconnected fault segments with different orientations, most of which are aligned with the main structural trend of the Plio-Quaternary extensional structures of*

*the Northern and Central Apennines (i.e. NW-SE; e.g., Galadini & Galli 2000; Barchi, 2010).*" This point is also addressed in the "Discussion" section and in the schematic reconstruction of the tectonic setting in Figure 16, where the analyzed faults are shown to dissect and displace the inherited faults, which are not reactivated contrarily to what stated in the text,.

**REPLY:** Once again, just to state it clearly, the main result from this work is the refinement of the architecture of the Monti Martani Fault System. In our view, this fault system is composed of NW-striking segments arranged in an en-echelon style. Its architecture is only influenced by the orientation of the inherited (Jurassic) structural grain (see also replies to the comments above). We never say in the text that the Plio-Quaternary extension reactivates the inherited faults. We also don't exclude (selective) reactivation of Jurassic faults during post-orogenic extension, but we have not specifically worked on this and so cannot explicitly document it. Instead, what we observed is that the most common expression of the post-orogenic expression along the MMR/Plio-Quaternary basins boundary corresponds to neo-formed, optimally oriented normal faults. This is clearly stated in the original manuscript (see, for example, lines 462-470).

This is quite confusing and the confusion about this topic is pervasive in the text, and in my opinion is related to the lack of field constraints that allow the Authors to infer if the NW-SE trending, en echelon normal faults they described played, or not, a crucial role in the structuration of the ridge.

**REPLY:** We do not understand this comment. We clearly state in several parts of the original manuscript that the post-orogenic NW-SE striking faults do not control the structuration of the ridge. We also state that the morphostructure of the ridge is controlled by the inherited structural grain, rather than by the Plio-Quaternary faults (see for example lines 19-22, 564-567).

One undeniable fact, in my opinion, is that the present-day morphostructural setting of the MMR is related to normal faults active at least in the Pliocene and the Early Pleistocene.

**REPLY:** Although the Reviewer is obviously entitled to his opinion, we are now supported by a wealth of original data that univocally suggest the newly proposed scenario, that is, that while the Plio-Quaternary faults strike NW-SE, the morphostructure of the ridge is L-shaped with N-S and WNW-ESE trending arms. This is what we reported in the original manuscript.

The post-orogenic activity of these normal faults, which may have followed original structural elements inherited from pre-orogenic tectonic phases (if not clear, this is a point that I support in this work, as I am confident that these faults played a significant role during the pre-orogenic phase), cannot be ignored because:

- these faults must have accommodated huge displacements. Evidence for this comes from the Cenozoic lithostratigraphic units that characterize the hanging walls of the faults, as observed both in outcrop (and some of these have even been measured on the field - see Viepri outcrops) and, more notably, in the subsurface. In contrast, Jurassic deposits predominantly occur at the footwalls of the master faults along the western and southern margins of the MMR. Despite these large displacements may result from polyphase tectonic activity, the fact that these faults downthrow Cenozoic carbonates and syn-orogenic Miocene deposits onto Jurassic units – such as those exposed at Grutti, few hundred metres north of the Viepri structural section- indicates that the minimum age of faulting is post-orogenic. Perhaps Similar features might exist in the subsurface of the Medio Tiberino and Terni basins, if not for the overlying Plio-Pleistocene deposits—otherwise, we wouldn't be debating this!

One suggestion: I invite you to take a look at the boreholes database (ex. Law 464/84 on the ISPRA website: https://sgi2.isprambiente.it/viewersgi2/). You could find useful informations;

**REPLY:** This is a very interesting point that sheds light on the long-known apparent discrepancy between the stratigraphic displacement that should have theoretically been accommodated by the Monti Martani Fault System and the lack of field structures (basically sufficient and sufficiently large faults) that could have accommodated such deformation. Barchi et al. (1991), based on field evidence and on a seismic section, already noticed the fundamental discrepancy between the total thickness of the Plio-Quaternary units (~500 m) and the cumulated displacement accommodated by the extensional fault system (thus, more than one fault!) bounding the western margin of the MMR (~2000 m). This discrepancy has not been solved yet and remains an open question in the geological history of the study area. By the way, several other examples of controversial younger-on-older relationships exist in the Northern and Central Apennines (e.g., Carminati et al., 2014; Calamita et al., 2017 and references therein).

A possible explanation is that the younger-on-older relationships that can be observed along basin-ward dipping faults or that can be reconstructed by integrating field evidence along the margins of the MMR with borehole data are not entirely attributable to post-orogenic extension. Similar relationships have been in fact interpreted as resulting from out-of-sequence thrusting or compressional reactivation of pre-thrusting extensional faults (e.g., Butler, 1989; Carminati et al., 2014; Calamita et al., 2017). Reverse kinematic indicators are indeed common in the vicinity of younger-on-older basin-ward dipping faults along the margins of the MMR. A noteworthy example (though not the only one) is located ~2 km to the NE of Castelrinaldi (north of Massa Martana), where a west-dipping high-angle fault separates the Scaglia Rossa in the hanging wall (to the W) from Jurassic limestone units in the footwall (to the E). Despite this younger-on-older relationship might suggest normal kinematics for this fault, the Scaglia Rossa contains very diffuse reverse top-to-the-E kinematic indicators, thus suggesting a compressional reactivation of a former extensional structure. This would imply that the stratigraphic gap observed across this structure cannot be exclusively attributed to post-orogenic extension. Of course, such an interpretation would require further studies to be validated, but considering the polyphase tectonic history of the study area it cannot be ruled out. For the moment, we do not have a large enough dataset to discuss the robustness of this interpretation, so we prefer not to discuss it in the present manuscript.

- the faults analyzed in this work do not justify the large stratigraphic displacement reported above, especially because they are intraformational or with limited downthrowing (with the exception of Viepri fault);

  **REPLY:** Please see the reply to the previous comment.

- the MMR-bounding faults controlled the deposition of the Plio-Quaternary deposits of the Medio Tiberino and Terni basins, as documented by published works on the Plio-Pleistocene deposits of these sectors (see Conti & Girotti, 1977; Ambrosetti et al., 1987, Basilici 1997), and as reported by the Authors themself in the text.

  **REPLY:** This is not correct. The Monti Martani Fault System somehow certainly steered deposition of (at least part of) the Plio-Quaternary infill of the Medio Tiberino and Terni Basins, but this does not imply that this tectonic control was exerted by a single major fault bounding the MMR.

  Additionally, the literature quoted by the Reviewer does not really state what he suggests. Indeed, Conti & Girotti (1977), when describing the bedrock/Plio-quaternary infill relationships, state: *"l'appoggio sulle strutture mesozoiche dei Monti Martani avverrebbe secondo una faglia; il limite occidentale di questi monti è infatti segnato da una grande faglia che borda la struttura da Nord a Sud, e della quale esistono chiari indizi in superficie, e che avrebbero offerto un*

*appoggio fortemente inclinato ai sedimenti lacustri*". They also talk about a "*grande faglia diretta che tronca la gamba occidentale dell'anticlinale mesozoica martana*". So, this work does not document the control of the N-S trending MMR-bounding fault. Like in most of the existing literature, the existence of this fault is only inferred, without offering any solid data/observation in support of it.

Likewise, concerning the role of the supposed Martana Fault, Ambrosetti et al. (1987) only say that the MMR is "*dominata da un motivo tettonico principale: la grande faglia bordiera che ha sbloccato l'anticlinale martana*". Again, this study does not contain any proof of the existence of the Martana Fault as a single, large displacement, N-S trending fault, as commonly accepted in the literature. This being the only statement on the Martana Fault in the entire article, Ambrosetti et al. (1987) do not document the control of this supposed fault on the Plio-Quaternary sedimentary fill of the basins.

Basilici (1997) says: "*the most important tectonic lineations bordering the Tiberino Basin are represented by NNW-SSE and, subordinately, ENE-WSW normal faults*". So, he is likely not referring to the "supposed" N-S and WNW-ESE fault(s) bounding the MMR.

Based on the above, some questions have arisen for me:

- If the activity of these faults was pre-orogenic, what produced the accomodation space for the accumulation of hundreds of metres (up to 2500 m) of marine-to-continental deposits during the Pliocene and Pleistocene?

  **REPLY:** At the risk of becoming repetitive, we need to yet again conclude that there is a misunderstanding. The Monti Martani Fault System, intended as a network of short and disconnected NW-SE striking faults distributed along the western and southern margin of the MMR (as we define it in the manuscript), did indeed play a role and likely produced the accommodation space for the sedimentary basins during the Plio-Quaternary. The accommodation space for the Plio-Quaternary units was likely produced by the cumulated displacement of all the post-orogenic NW-SE striking normal faults documented along the western and southern margins of the MMR.

  However, some clarifications are needed. The Plio-Quaternary succession of the Medio Tiberino and Terni Basins is continental, not marine-to-continental (see Ambrosetti et al., 1987; Basilici, 1992, 1997; Bonini et al., 2003). Also, the total thickness of the sedimentary infill of these basins is likely in the order of 500 m, rather than "up to 2500 m". Estimates of such thickness made by adding the thickness of each sedimentary unit are of about 500 m (e.g., Basilici, 1997) and agree with the information derived from seismic reflection profiles (Barchi et al., 1991). The idea that the sedimentary succession of these basins could be thicker than this derives from a gravimetric study performed by Ambrosetti et al. (1993) that propose a maximum thickness of 2300 m in the Collevalenza area (southwest of Massa Martana). Since this study is only referred to in the literature and it is impossible to retrieve it (it has been published in the proceedings of the progress meeting of an international project), it is not possible to judge the quality of the data and the solidity of their interpretation. However, Valentini et al. (1997) proposed a reasonable solution to this discrepancy. Referring to Ambrosetti et al. (1993)'s work, they propose that "*it is possible that differences in the thickness (…) have been emphasized by schematic modeling of the substratum. In fact, if intermediate density "strata", such as turbidite and pre-turbidite marly successions, are introduced into the gravity model, the total thickness of the Plio-Quaternary sediments would be significantly reduced*" (Valentini et al., 1997).

Thus, it is much more likely that the total maximum thickness of the Plio-Quaternary infill of the basins is about 500 m.

- What produced the erosional surface on which the Plio-Quaternary marine-to-continental successions rest on?

    **REPLY:** This is an interesting question, but we do not have data to solve it, and it is in any case far beyond the scope of our study.

- Why the N-S segment couldn't have acted as transfer fault of NW-SE o WNW-ESE faults, such those occurring North of Martano Mt and along the southern border of the MMR?

**REPLY:** This is an interesting point. However, our structural recognition along the border of the MMR does not reveal evidence of recurrent N-S segments (except for site 5, for which dip-slip or oblique movements are documented; see the stereoplots we provided in the in manuscript). As discussed throughout this rebuttal letter, NW-SE-striking faults are the surficial expression of the post-orogenic setting, not the N-S-striking ones.

- Could the faults analysed in this work be the result of strain partitioning related to the activity of the faults bordering the MMR rather than faults related to a different genetic process (i.e., a different, Quaternary, stress field)? In fact, the whole analysed faults, except for the Sangemini structural stations, occur at the footwall of the master faults, that now are buried by Quaternary deposits beneath the Terni and Medio Tiberino basins, and whose kinematics are not clear.

**REPLY:** If the NW-SE-striking faults were the result of strain partitioning during the activity of the fault bordering (i.e., the N-S-striking, we imagine) the MMR, it would be evident that N-S-striking faults are the dominant set, while the NW-SE-striking faults are subordinate. However, as stated above, this is not the case, as our findings align with the data existing in the literature (i.e., Barchi et al., 1991; Brozzetti & Lavecchia, 1995; Bonini et al., 2003).

These are issues that are not discussed in your paper and which, in my opinion, need to be addressed and discussed.

**Role of post-rift and pre-orogenic tectonic phases in the Umbria-Marche Sabina area**

The Apennines experienced polyphase tectonics, and this has impacted in the structural and stratigraphic evolution of this range. Under-estimation of post-Tethyan rift and pre-orogenic tectonic deformations could have repercussions on the structural analysis of selected areas. There is a growing literature considering synsedimentary extension-dominated tectonic phases affecting the Apennines in the post-rift to syn-orogenic time span (e.g., Bajocian, Barremian-Aptian, Cenomanian, Maastrichtian, Paleocene, Miocene - see, for istance, Centamore et al., 2007; Cipriani & Bottini, 2019a,b; Capotorti & Muraro, 2021, 2024; Sabbatino et al., 2021). The latter should have repercussions on the inferences discussed in this work, and cannot be underestimated.

I suggest to introduce at least in the "Geological setting" section, information about the occurrence of these tectonic phases.

**REPLY:** We thank the reviewer for this suggestion. We will add this information in the Geological Setting in the revised version of the manuscript.

**Relationships between carbonate bedrock and Plio-Quaternary deposits**

In the lines 405-408 is written: "... *the contact between the Upper Pleistocene-Holocene deposits of the Medio Tiberino and Terni basins and the Meso-Cenozoic carbonate and siliciclastic rocks of the MMR is an unconformable stratigraphic contact and not an extensional tectonic contact as proposed by earlier studies (e.g., Ambrosetti et al., 1987; Bonini et al., 2003*)."

Assuming that one does not exclude the other, I agree with the fact that at present the contact between the upper Quaternary deposits and the carbonate bedrock is of a stratigraphic and unconformable nature, but this cannot be the case for the Pliocene and Lower Pleistocene deposits (which are not exposed along the western edge of the MMR) and the Meso-Cenozoic rocks. That said, the present unconformity surface could be the legacy of the Late Pliocene and Early Pleistocene faulting whose fault scarps, similarly to what happened for the Early Jurassic faults, have been then shaped by morphogenetic agents and become the site of stratigraphic and no longer tectonic contacts. Comparable features have been documented in the close Narni-Amelia Ridge by Cipriani (2016, 2019); here, Pliocene faults are buried by, and fault scarps are onlapped by, marine to continental Plio-Quaternary deposits. Furthermore, the fault scarps are overprinted by bioerosions (lithophagous holes).

As a consequence, these stratigraphic relationships cannot permit to exclude a Plio-Pleistocene activity for the Martana Fault.

**REPLY:** We agree with the Reviewer, and in fact, as it is also written in the part of the manuscript quoted by them, here we are just referring to the relationship between the fault system and the Upper Pleistocene-Holocene deposits. Nowhere in the manuscript we exclude the activity of the Monti Martani Fault System during the Late Pliocene-Early Pleistocene. By contrast, our work would highlight that the idea that subsidence in the Middle Tiberino and Terni Basins was manly controlled by a supposed N-S and WNW-ESE trending Martana Fault bounding the MMR is not supported by any data. Our study shows that the majority of the extensional faults outcropping along the western and southern margins of the MMR strike NW-SE and there is no solid data in the existing literature proving the activity of major N-S or WNW-ESE faults during the opening of the Medio Tiberino and Terni basins. Even if one looks at the existing data concerning extensional faults that cut the older units (Upper Pliocene-Lower Pleistocene) of the sedimentary fill of these basins (see Basilici, 1992, 1997; Barchi et al., 1991; Brozzetti & Lavecchia, 1995; our work), these all show that the vast majority of the post-orogenic faults strike ~NW-SE (i.e. optimally oriented with respect to the regional post-orogenic extensional stress field). Thus, we do not understand why the Reviewer perceives this part of the discussion as inconsistent.

Concerning the stratigraphic displacement that should have been accommodated by the Monti Martani Fault System during the post-orogenic extension, please see our replies to previous comments.

Another problem is the age provided for Plio-Quaternary deposits. I do not believe, in fact, that the purpose of this work was to provide, and therefore that you derived, the age of the deposits. In my opinion, it is necessary to quote the references the Authors used to provide the age of Plio-Quaternary deposits (see, for istance, Conti & Girotti, 1977, Ambrosetti et al., 1978, 1987; Basilici,

1997), introducing a sentence in the text where they explain why have been decided to rely on one author rather than another. This topic has inferences on the following points of discussion.

**REPLY:** We thank the Reviewer for this suggestion and we will add this information in the revised version of the manuscript. Just to clarify it also in this response, for the age of the Plio-Quaternary units of the Medio Tiberino and Terni Basins that we described in this work we referred to the stratigraphic schemes of Basilici (1997), Bonini et al. (2003) and the 1:10.000 scale geological map of the Regione Umbria. All these works agree on the age of the deposits described in the San Gemini area and along the western and southern margins of the MMR.

*Structural stations and reconstruction of tectonic phases relationships*

Punctual structural data cannot allow the Authors to confirm nor the reactivation in the post-orogenic phase of pre-orogenic structures, nor to limit their activity to the pre-orogenic stage, nor to discriminate if the studied faults dissect the inherited ones. Inferences about the relationships between tectonic elements can be made if the relative age of the activity of each fault is defined. Except for the structural stations of Viepri and Cesi, most of the analysed faults are intraformational and, even the identification of polyphase reactivations, a relative chronology for each phase of tectonic activity cannot be constrained. This is especially valid when analysing faults (and fractures) affecting the Calcare Massiccio. In fact, I want to emphasise the fact that the Calcare Massiccio has recorded all the various deformation phases (whether more or less important) that have involved today's north-central Apennines, from the Tethyan rifting to post-rift and pre-orogenic (Bajocian, Barremian-Aptian, Maastrichtian, Paleocene, Miocene) normal faulting, to synorogenic compression to post-orogenic extension. Taking into account this information, it would be important to recognise and provide a relative chronology of structures, that could be identified not in limited structural stations, but after a widespread geological mapping involving the whole area.

**REPLY:** It is of course true that we collected data at punctual structural stations (as it is always the case in structural geology studies that use similar approaches), but these locations are not scattered spots randomly distributed in the study area. Rather, the studied outcrops are located along the boundary between the MMR and the adjacent Plio-Quaternary basins, where we carried out systematic analyses in order to establish the structural relationships between the ridge and the sedimentary infill of the basins (which was the objective of our study, as said before). In this view, we find quite reductive to define the dataset presented in this work as "punctual structural data".

As noticed by the Reviewer, it is true that we did not document any crosscutting relationship between the faults described in this work and pre-existing structures possibly related to the pre-orogenic tectonic history. A "widespread geological mapping" exercise of the whole region would certainly bring new constrains on this matter, although the lack thereof does not imply that reasonable, well documented and scientifically sound interpretations cannot be proposed. These extrapolations are in fact possible because our observations are systematically and not randomly located.

We are aware that the Calcare Massicio has undergone a polyphase tectonic history and may have recorded all the tectonic events that occurred between the Early Jurassic rifting to the Plio-Quaternary post-orogenic extension. We are also aware that it is not always easy to attribute a particular structure in the Calcare Massiccio (in any rock type, for that matter) to any of these phases if proper geological constraints are missing. This is exactly why we did not use the fault data from the structural station in the Acquasparta quarry (station 3 in Fig. 3 and stereonet 3 in Fig. 6 of

the manuscript) to perform tensorial analysis. However, in other cases, it was possible to establish the relative chronology between different tectonics events. These cases were used as pinpoints for reasonable extrapolations to the other structural stations where similar constraints are missing. For example, in the Viepri site (site 1 in Fig. 3 of the manuscript) the described normal fault juxtaposes the Lower Jurassic Corniola in the footwall with the Eocene Scaglia Variegata in the hanging wall. Since there is no extensional event younger than the Eocene documented in the region, the extensional activity of this structure can be reasonably attributed to the post-orogenic extension. This is in agreement with previous interpretations of the same fault (Barchi et al., 1991; Brozzetti & Lavecchia, 1995; Bonini et al., 2003). Similarly, in the Grotta Eolia site in Cesi (site 8 in Fig. 3 of the manuscript) a normal fault juxtaposing the Calcare Massiccio in the footwall against the Rosso Ammonitico (?) in the hanging wall. Here, as stated in the manuscript, fault planes related to the syn-orogenic phase are also present, but they are crosscut by the extensional structures. For this reason, not all the fault planes observed in this site were used to perform tensorial analysis. Thus, also at this site, extensional structures can only be attributed to the post-orogenic phase.

All the other extensional structures that we have documented in this study are oriented and/or have a slip direction that is comparable to those of the sites mentioned above. Furthermore, the tensorial analysis performed on these structures retrieved results (i.e., NE-SW extension) that are comparable to those performed in the Viepri and Grotta Eolia sites and also on extensional faults cutting through the Late Pliocene-Early Pleistocene units of the sedimentary fill of the Medio Tiberino Basin, that are certainly related to the post-orogenic extension (Barchi et al., 1991; Brozzetti & Lavecchia, 1995; this study). For all these reasons, we find more than reasonable to attribute the faults we used to perform tensorial analysis to the post-orogenic extensional event.

A general comment, moreover, needs to be made as to the technique of stress inversion and sorting of fault-slip data collected in the field. If we were to systematically exclude all intraformational faults, we could easily set aside the technique of stress inversion in this kind of geological setting. Although direct cutting relationships are indisputably the only evidence (except for direct dating of structural features) that confidently help us establish the relative timing of faulting episodes and, therefore, only faults that cut across stratigraphic contacts bear direct implications on the age of faulting relative to the age of the younger formation cut across, there are indeed many other criteria that are used to sort complex fault-slip data. Geometrical, kinematic and mechanical compatibility criteria, average size of striated fault planes, mineral coatings, etc., etc., are all conceptual tools that we collect in the field and that we carefully and systematically use when computing paleostress tensors. By a systematic approach based on such a multi-faceted approach we can confidently assign faults to a specific stress tensor solution or remove them from a given dataset, even if those faults are found to be intraformational. Suggesting to remove intraformational faults is thus not tenable.

**Seismic hazard**

About the seismic hazard and the lenght of the faults analysed, in my opinion the data shown and discussed in this paper are not sufficient to assume that these faults are not seismogenic and that has no high potential magnitude. In order to infer the latter, you have to map the whole length of the faults and discriminate that those are limited segments (at the surface!). In fact, why they could not be linked at depth, producing tens of km-long faults? Palaeoseismologists could comment on the work by inviting you to do, for example, further investigations such as palaeoseismological trenches etc. This point needs of further field data.

Moreover, you have no geochronologic constraints for determining the age of dissected or sealing Quaternary deposits. Few thousand years can move the needle of the scale towards the capability of these lineaments to dissect the topographic surface.

**REPLY:** In Section 5.4 of the manuscript, we state that "the distribution of active seismicity suggests that the MMR is a seismically active region that responds to the extensional tectonic regime currently shaping the internal domain of the Apennines", and we also illustrate instrumental and historical seismicity in the study area in Fig. 2. With our study, we only question the definition of the fault system as active and capable, since we did not find any evidence of fault surfaces cutting through deposits younger than Lower Pleistocene, nor this evidence is presented in the literature.

Considering the geometric relationship between the faults described in this study at the western and southern margins of the MMR (i.e., en-echelon), it is very unlikely that they link at depth to produce a tens of kilometers-long structure, because they are misaligned. Furthermore, it is very unlikely that structures of such a size lack a clear (and yet known) surface expression. The complete mapping of these faults would be for sure a more solid constraint to the model we propose. However, these structures are very hard to follow along-strike and their complete mapping is often impossible, thus suggesting they do not represent a mature, well-linked structural system (unlike the Vettore Fault or the Morrone Fault of the Central Apennines). Basin-ward, these NW-SE faults are covered by Upper Pleistocene units or their exposure is not preserved due to the extremely bad outcrop conditions of the Plio-Quaternary continental units. Toward the ridge, they get soon lost in the vegetation or in hardly accessible areas. However, the fact that existing geological maps and the geological cartography of Regione Umbria at 1:10,000 scale representing the internal part of the ridge do not show the presence of major NW-SE striking faults, is in our opinion supportive of the idea that these faults have short lengths and are mainly distributed along the ridge/basins boundary. This allows us to infer that their seismogenic potential is relatively low, as also supported by the historical and instrumental seismicity in the study area (see Fig. 2 of the manuscript). This is the first-order structural approach that is commonly used to assess the seismic hazard of faults in the field.

We appreciate that the Reviewer would like us to do every possible scientific investigation in the Martani area, but not everything can be done in the frame of a single study. Paleoseismological trenching is an art in its own and requires expertise, experience and funds. Moreover, we do not think that it would bring any further constraints on this subject since the faults that we have documented to be sealed by Upper Pleistocene units cut through the Meso-Cenozoic bedrock or the Upper Pliocene-Lower Pleistocene units. Thus, it is very unlikely that this kind of studies would bring constraints on the recent activity (< 40 kyrs) of these faults.

Concerning the age of the dissected or sealing Quaternary deposits described in this work, it is true that we did not provide any new geochronological constraints. However, we referred to published chronostratigraphic schemes and geochronological attributions that are generally accepted in the region, such as Basilici (1977), Bonini et al. (2003) and the Regione Umbria 1:10.000 geological map. All these works attribute the same ages to the Plio-Quaternary units described in this work (i.e., Upper Pliocene-Lower Pleistocene for the faulted units; Upper Pleistocene for the sealing units).

While we are confident in the logic and reasonableness of our conclusions, we acknowledge that they are partially based on assumptions that lack of absolute constraints (like, for example, the age of the sealing Upper Pleistocene units). This is why in this part of the manuscript we do not use an affirmative language (e.g., proving, demonstrating, etc.), but we only say that, based on these

considerations and the observations presented in our study, the definition of the Monti Martani Fault System as active and capable "should be reconsidered".

*Paleostress evolution*

Another major weakness points of this work that should be addressed are represented by the reconstruction of the extensional tensors. In fact, the three identified orientations of σ3 (i.e, NE-SW, NNE-SSW and NW-SE) cannot be chronologically constrained, as yourself stated in lines 480-481. Despite that, you implicitly infer that the NW-SE oriented σ3 is post-orogenic. Based on what? Furthermore, why infer the existence of a further post-Pliocene but pre-Late Pleistocene NW-SE extensional deformative phase, for which there is no direct evidence (e.g. inherited faults displaced by younger NE-SW extensional faults)? Would it not be more parsimonious to link their presence to differential movements (i.e. strain partitioning) at the footwall of the faulted blocks, assuming the existence of buried master faults? Sorry but it is not clear to me.

**REPLY:** In lines 480-481 we do not say that the three identified paleostress tensors cannot be chronologically constrained. There, we wrote that "*field observations did not allow us to establish the relative chronology between the two first extensional phases*" (i.e., NE-SW and NNE-SSW extension directions; lines 475-476 of the original version). It does not mean that we do not have reasonable arguments to attribute these extensional paleostress fields to the post-orogenic extension. For what concerns the evidence in support of the attribution of the NE-SW and NNE-SSW extensional phases to the post-orogenic regime, please see our reply to the comment on "Structural stations and reconstruction of tectonic phases relationships". Since field evidence shows that the structures related to the NW-SE oriented σ3 clearly postdate the orogen-orthogonal extension (see for example Figs. 8 and 12 in the manuscript), this paleostress tensor can only be framed in the post-orogenic phase. For example, if one looks at the Viepri site (Fig. 8 in the manuscript), the NW-SE oriented σ3 was reconstructed based on veins cutting through (and thus postdating) the damage zone of a NW-SE striking normal fault. Since this fault bears Eocene units in its hanging wall, its activity can only be syn-to-post-orogenic, so as every structure post-dating it. This example shows that it is not true that there is no direct evidence for a NW-SE extensional deformation phase.

Concerning the tectonic/geodynamic interpretation of this latter extensional phase, we agree with the Reviewer that the way we discussed it in the manuscript was somehow misleading. This point has been in fact also risen by the other Reviewer (Marco Mercuri). Our intention was not to necessarily attribute this deformation phase to a regional geodynamic event. In fact, in the manuscript we state that "*the geodynamic causes of this orogen-parallel extensional event remain unclear*". Afterwards in the manuscript, we discuss the existence of similar tectonic scenarios (i.e., orogen-parallel and orogen-orthogonal extension in the frame of post-orogenic extension) with the relative proposed interpretations, since we find interesting to discuss analogue situations even though a unique interpretation of such settings does not exist at present. We also highlight the possible synchronicity of these orogen-parallel extensional event with a phase of regional uplift (see lines 505-506 in the manuscript), but without necessarily claiming for a genetic link between the two events. We are aware that the explanation for the existence of structures related to an orogen-parallel extensional paleostress tensor might also depend on local perturbations of the regional stress field in mature stages of fault evolution (as also pointed out by Reviewer #1), and that the way we discussed these structures in the manuscript might suggest that we favor a regional tectonics explanation. For this reason, in the revised version of the manuscript, we will add references to the possibility that orogen-parallel extensional paleostress tensor might also result

from local perturbations of the stress field during fault evolution (see also the reply to the comment on the Discussion section in the response to Reviewer #1).

**Technical corrections**

*Geological setting*

Except for the "*formazione marnoso-arenacea*", all the lithostratigraphic units for the Umbria-Marche succession do not have the term "formation" in their name, and I suggest to delete "Fm." from each unit.

**REPLY:** We thank the Reviewer for this suggestion that we will integrate in the revised version of the manuscript.

Line 79: Please do not use the term 'series', which is a formal chronostratigraphic unit, as a synonym for 'succession'. I suggest replacing "series" with "succession" in the text.

**REPLY:** We thank the Reviewer for this suggestion that we will integrate in the revised version of the manuscript.

Line 110: "Tertiary" is no more a chronostratigraphic unit, and should be changed with Cenozoic.

**REPLY:** We thank the Reviewer for this suggestion that we will integrate in the revised version of the manuscript.

Lines 111-113 : I suggest to make reference to a wider literature (i.e., Calamita & Pierantoni, 1994, 1995; Alfonsi et al., 1991, 1995)

**REPLY:** We thank the Reviewer for this suggestion that we will integrate in the revised version of the manuscript.

Line 117: Works that were focused on the Jurassic deposits are not quoted here. I suggest to introduce at least these references: Mariotti et al, 1979; Farinacci et al., 1981; Galluzzo & Santantonio, 2002

**REPLY:** We thank the Reviewer for this suggestion that we will integrate in the revised version of the manuscript.

Line 120: The faults described by Bruni et al. are submarine fault palaeoscarpments. Before they were passively buried by the hangin wall basins filling successions and thus became an unconformity surface rather than a fault surface, these escarpments have undergone a complex evolution in their geomorphic evolution, which may have led them to assume geometries inconsistent with those of the rooted faults that generated them. Gravitational retreat is the main examples. For more details see Carminati & Santantonio, 2005 and Santantonio et al., 2017. As a consequence, I suggest caution in relying on this kind of data from a structural point of view.

**REPLY:** We thank the Reviewer for raising this point. We will modify this sentence in the revised version of the manuscript in order to better clarify these uncertainties.

Lines 148-149: "*The Medio Tiberino Basin is a N-S to NNW-SSE trending graben to the west of the MMR and is filled by a ~500 m thick Upper Pliocene-Quaternary continental succession (Barchi et al., 1991; Basilici, 1997).*" The thickness of these deposits reach up to 2300 m in the Collevalenza area (Ambrosetti et al., 1993 in Basilici, 1997).

**REPLY:** Please see the reply to the previous comment on "Role of Jurassic rifting faults and present-day geometry".

In lines 160-170 the reference to Fig. 1b is erroneous, I suppose. Maybe you refer to Fig. 2 and Fig. 3.

**REPLY:** Yes, indeed, the reference to Fig. 1b is wrong here. The correct reference was to Fig. 2. We will correct in the revised version of the manuscript.

**Methods**

Which kind of software have you used for the realization of stereographic projections? Please, provide a reference.

**REPLY:** For stereographic projections we used the Win-Tensor program (Delvaux, 1993), which is the same we used for paleostress analysis. We will add this information in the revised version of the manuscript.

**Discussion**

Lines 425-428: I believe that in these reference lists (and in the geological setting section as well)important works focussed on the structural relationships of the Jurassic inheritances on the orogenic deformations, also involving the Martani Mts, are missing, as: Calamita & Pierantoni, 1994,1995; Scisciani, 2009, Scisciani et al., 2014; Curzi et al., 2024.

**REPLY:** We thank the Reviewer for this suggestion that we will integrate in the revised version of the manuscript.

**Figures**

**Figure 2:** I suggest to update this figure introducing the main structural elements lacking in the present form, as drawned. I suggest to see the structural scheme by Calamita & Pierantoni (1995).

**REPLY:** We thank the Reviewer for this suggestion. We will modify the figure accordingly.

**Figure 3:** see previous comments. Furthermore, in the legend you talk about deposits. Consequently, chonostratigraphic rather than geochronologic units should be used.

**REPLY:** We thank the Reviewer for this suggestion. We will modify the figure accordingly.

**Figure 4:** see previous comments. Furthermore, Lias, Dogger and Malm are no more chronostratigraphic units, and should be avoided. Please, change with Lower, Middle and Upper.

**REPLY:** We thank the Reviewer for this suggestion. We will modify the figure accordingly.

**Figure 7:** Abbreviations in the triangular schemes should be provided in the caption.

**REPLY:** We thank the Reviewer for this suggestion. We will modify the figure accordingly.

**Figure 11:** *Neptunian dike*: as the name itself suggests, "neptunian" means that it was formed under water. Those in the figure are "clastic dikes", or "fissures filled with clastic, continental sediments".

**REPLY:** We thank the Reviewer for pointing out our inappropriate use of the term "neptunian dike". We will replace it in the figure and the manuscript with the more appropriate "fissure infill".

**Figure 12:** in panels a and c the strike slip slickenlines are not so clear.

**REPLY:** In the revised version of the manuscript we will increase the resolution of this figure in order to make the strike slip slickenlines more visible.

**Figure 13:** I'm not sure that what you indicate in this figure as "fractured bedrock" is really fractured bedrock. In fact, looking at the deposits it seems to be heterometric breccias, but above all chaotic (and not with a fitted fabric type structure that I would expect for a fractured substrate) with a finer reddened matrix. Are you sure that they are not continental deposits older than those indicated as "Upper Pleistocene slope debris"?

**REPLY:** As also stated in the same figure, we cannot exclude a partial remobilization of these clasts. This is why this part of the outcrop is labeled as "*(partially remobilized?) brecciated bedrock*". We agree with the Reviewer that this nomenclature is more appropriate than "*fractured bedrock*", so we will replace it in the revised version of this figure.

This feature has also been observed elsewhere along the margin of the MMR. It consists of a monomictic, angular, heterometric breccia whose thickness does not exceed 2-3 m. The composition of the clasts is the same as that of the bedrock it rests on and the fabric is chaotic. For these reasons we interpret it as resulting from local (in situ) brecciation of the bedrock without excluding a limited remobilization of the clasts.

For the sake of clarity, this information will be added and better explicated in the revised version of the manuscript.

**Figure 16:** This figure is not correct. A W-dipping normal/trastensive fault displaced by SW-dipping extensional faults should have its trace dissected and shifted towards E in plant view, not towards W (see line drawing on the pdf).

**REPLY:** We thank the reviewer for noticing this inconsistency. The figure will be corrected accordingly.

Moreover, it is oversimplified. In fact, the southern boundary extension structure of the MMR is not oriented E-W, but WNW-ESE, and thus has an angle of about 20° of deviation from that of the regional sigma3.

**REPLY:** This figure is meant to present a conceptual model (as stated in the caption) and, for this reason, it is simplified. The strike of the southern boundary of the MMR is ~N100 (e.g., Brozzetti & Lavecchia, 1995; Bonini et al., 2003). Thus, the angle with the regional $\sigma_3$ is about 35°, not 20°. Anyway, we will modify this figure representing a more accurate and realistic trend of this margin of the MMR.

**References**

Ambrosetti, P., Carboni, M. G., Conti, M. A., Esu, D., Girotti, O., La Monica, G. B., ... & Parisi, G.: Il Pliocene ed il Pleistocene inferiore del bacino del Fiume Tevere nell'Umbria meridionale: The Pliocene and the Lower Pleistocene of the Tevere Basin in Southern Umbria. Geografia Fisica e Dinamica Quaternaria, 10(1), 10-33, 1987.

Ambrosetti, IZ, Barbieri, M., Basilici, G., Bozzano, L, De Pari, E, Di Filippo, M., Di Maio, R., Duddridge, (i., Etiope, G., Gambino, E, Graingcr, P., Lombardi, S., Mottana, A., Patella, D., Pennacchioni, E., Ruspandini, T., Scarascia Mugnozza, G., Sordoni, G., Tazioli, S., Toro, B., Valentini, G. and Zuppi, G.: Analysis of geoenvironmental conditions as morphological evolution factors of the sand-clay wries of the Tiberino valley and Dunarobba Forest preservation. Proceedings of the Progress Meeting on the Mirage project, 3rd phase, Bruxelles, 7-8 October 1993, 17 pp, 1993.

Barchi, M., Brozzetti, F., and Lavecchia, G.: Analisi strutturale e geometria dei bacini della media valle del Tevere e della Valle Umbra. Bollettino della Società Geologica Italiana, 110, 65–76, 1991.

Basilici, G.: Sedimentary facies in an extensional and deep-lacustrine depositional system: the Pliocene Tiberino Basin, Central Italy. Sedimentary Geology, 109, 73–94, 1997.

Bonini, M.: Chronology of deformation and analogue modelling of the Plio-Pleistocene 'Tiber Basin': implications for the evolution of the Northern Apennines (Italy). Tectonophysics, 285, 147–165, 1998.

Bonini, M., Tanini, C., Moratti, G., Piccardi, L., and Sani, F.: Geological and archaeological evidence of active faulting on the Martana Fault (Umbria-Marche Apennines, Italy) and its geodynamic implications. J. Quat. Sci., 18, 695–708, 2003.

Brozzetti, F., and Lavecchia, G.: Evoluzione del campo degli sforzi e storia deformativa nell'area dei Monti Martani (Umbria). Bollettino della Società Geologica Italiana, 114, 155–176, 1995.

Butler, R.W.H.: The influence of pre-existing basin structure on thrust system evolution in the Western Alps. In: Inversion Tectonics (M.A. Cooper & G.D. Williams, eds). Geological Society, London, Special Publications, 44, 105–122, 1989.

Calamita, F., Di Domenica, A., and Pace P.: Macro- and meso-scale structural criteria for identifying pre-thrusting normal faults within foreland fold-and-thrust belts: Insights from the Central-Northern Apennines (Italy). Terra Nova, 30, 50–62, 2017.

Carminati, E., Fabbi, S., and Santantonio, M.: Slab bending, syn-subduction normal faulting, and out-of-sequence thrusting in the Central Apennines. Tectonics, 33, 530–55, https://doi.org/10.1002/2013TC003386, 2014.

Conti, M. A., and Girotti, O.: Il Villafranchiano del" lago Tiberino", ramo Sud occidentale. Schema stratigrafico e tettonico. Geologica Romana, 16, 67-80, 1978.

Giglia, G., Ronga, G., and Trevisan, L.: Idrogeologia della zona di Sangemini. Collana scientifica Centro Studi Sangemini, 3, 11-30, 1977.

Valentini G., Lombardi S., Bozzano F., and Scarascia Mugnozza G.: Analysis of the geo-environmental conditions as morphological evolution factors of the sand-clay series of the Tiber valley and Dunarobba forest preservation. European Commission, nuclear science and technology, Report EUR 17479, ISSN 1018-5593, 1997.

---

## Author Comment (AC3)

[revised manuscript text omitted]

---

## Author Response (AR2)

*Dear Editor Federico Rossetti,*

*We wish to thank you for the editorial handling of our manuscript entitled "Reconciling post-orogenic faulting, paleostress evolution and structural inheritance in the seismogenic Northern Apennines (Italy): Insights from the Monti Martani Fault System". Comments and suggestions by the Topic Editor Stefano Tavani have been thoroughly considered and integrated into the revised version of the manuscript. This surely helped us to improve the quality of the article.*

*In the following, we respond point-by-point to each of the Topic Editor's comments. We report the original comments in* **black***, followed by the relative responses in* **red***. We are confident that the suggested adjustments will strengthen the overall quality of the manuscript and we hope that our replies to the Topic Editor's comments and the way we integrated them in the revised version of the manuscript will meet your satisfaction.*

*Yours sincerely,*

*Riccardo Asti*

*(on behalf of the co-authors)*

**RESPONSE TO TOPIC EDITOR (Stefano Tavani)**

**Original comments:**

The manuscript is now well-structured and adequately addresses the reviewers' comments. I have just a few technical requests:

Lines 59 to 61: Not entirely correct. The NE portion of the central Apennines belongs to the Umbria-Marche domain of the northern Apennines, and the Apennine platform of the southern Apennines is simply the southern continuation of the Lazio-Abruzzi platform from the central Apennines. Remove the sentence.

**REPLY:** this sentence has been removed.

Please remove the Ortona-Roccamonfina line from both the text and Figure 1a. Its existence, initially proposed in early studies based on satellite lineaments, is at best questionable.

**REPLY:** references to the Ortona-Roccamonfina tectonic line have been removed from both the text and Figure 1a.

Reinsert "Fm." after the names of all formations and remove "Auctt." While these terms are formally correct in Italian nomenclature, without "Fm." readers outside of central Italy would not understand that "Calcare Massiccio" refers to a formation.

**REPLY:** these modifications have been made in the text and in Figure 4.

Replace "structural high" and "structural low" with "horst" and "graben".

**REPLY:** these modifications have been made in the text and in Figure 4.

Move Table 1 to the supplementary material.

**REPLY:** this table has been moved to the Supplementary Material.